# Classifying the CP properties of the ggH coupling in H + 2j production

Henning Bahl[1⋆], Elina Fuchs[2,3,4†], Marc Hannig[3‡] and Marco Menen[3,4∘]

**1** University of Chicago, Department of Physics,
5720 South Ellis Avenue, Chicago, IL 60637 USA
**2** CERN, Department of Theoretical Physics, 1211 Geneve 23, Switzerland
**3** Institut für Theoretische Physik, Leibniz Universität Hannover,
Appelstraße 2, 30167 Hannover, Germany
**4** Physikalisch-Technische Bundesanstalt, Bundesallee 100,
38116 Braunschweig, Germany

⋆ hbahl@uchicago.edu , † elina.fuchs@cern.ch ,
‡ marc.hannig@stud.uni-hannover.de , ∘ marco.menen@itp.uni-hannover.de

## Abstract

The Higgs-gluon interaction is crucial for LHC phenomenology. To improve the constraints on the $CP$ structure of this coupling, we investigate Higgs production with two jets using machine learning. In particular, we exploit the $CP$ sensitivity of the so far neglected phase space region that differs from the typical vector boson fusion-like kinematics. Our results indicate that further improvements in the current experimental limits could be achievable using our techniques. We also discuss the most relevant observables and how $CP$ violation in the Higgs–gluon interaction can be disentangled from $CP$ violation in the interaction between the Higgs boson and massive vector bosons. Assuming the absence of $CP$-violating Higgs interactions with coloured beyond-the-Standard-Model states, our projected limits on a $CP$-violating top-Yukawa coupling are competitive with more direct probes like top-associated Higgs production and limits from a global fit.

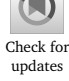

# 1   Introduction

More than ten years ago, the ATLAS and CMS collaborations announced the discovery of a new particle at a mass of about 125 GeV during Run-1 of the Large Hadron Collider (LHC) [1, 2]. Since its discovery in 2012, the quantum numbers of this Higgs boson have been tested extensively, as well as its interactions with other SM particles [3–6]. Up to now, all results are in agreement with the predictions of the SM within the experimental and theoretical uncertainties. However, effects from beyond the Standard Model (BSM) physics could still be hidden in the current uncertainties and may be unveiled with the large amount of data to be collected during LHC Run-3 and the high-luminosity phase of the LHC (HL-LHC).

A so far relatively unconstrained property of the discovered Higgs boson is its behaviour under $\mathcal{CP}$ transformations. This is of particular interest given that the amount of $\mathcal{CP}$ violation present in the SM is — by several orders of magnitude — insufficient to explain the baryon asymmetry of the Universe [7, 8]. The possibility of the Higgs boson being a pure $\mathcal{CP}$-odd state was already ruled out shortly after its discovery by analyzing the $\mathcal{CP}$ structure of its couplings to massive vector bosons [3, 4]. It is, nevertheless, still possible that the structure of the discovered Higgs boson's couplings is $\mathcal{CP}$-mixed instead of (nearly) $\mathcal{CP}$-even as predicted by the SM.

Electric dipole moments (EDMs) are sensitive probes of $\mathcal{CP}$ violation beyond the SM [9,10] and their experimental upper bounds, in particular of the electron [11, 12], neutron [13] and mercury [14], place strong constraints on $\mathcal{CP}$-violating Higgs interactions [15–19]. These constraints, however, strongly depend on the assumption about the first-generation Yukawa couplings [20], which themselves are only very weakly constrained [21–27, 27–30]. It is therefore of great interest to search for possible $\mathcal{CP}$-violating effects also at colliders, which allow for a distinction between different Higgs couplings.

At the LHC, $CP$-violating Higgs interactions can be constrained using the kinematic information of the final state particles. Most of the observables built from these kinematics are $CP$-even, but can show sensitivity towards $CP$ violating effects via changes in their distributions. However, an observed deviation in a $CP$-even observable is only indicative of $CP$ violation. Dedicated $CP$-odd observables offer an advantage as measuring a non-zero value for them is an unambiguous sign of $CP$ violation. On the other hand, it can be difficult to measure $CP$-odd observables since it often requires measuring at least four independent momenta associated either with the Higgs production or with its decay (see e.g. [31]). In this situation, the measurement of $CP$-sensitive observables can provide valuable complementary information.

The most stringent constraints on $CP$-violating Higgs couplings so far have been set on the Higgs interactions with massive vector bosons [3, 4, 32–41]. In most BSM theories, $CP$ violation in these interactions is expected to be loop-suppressed (given that a pseudoscalar cannot be coupled to massive vector bosons at the tree level). $CP$ violation in the Higgs interactions with fermions can, in contrast, occur unsuppressed. These are far less constrained with the existing experimental analyses targeting the Higgs interaction with tau leptons [42, 43] and with top quarks [39, 44–49].

Since the top-Yukawa coupling is of special interest due to its magnitude, also many phenomenological studies have been carried out focusing mainly on top-associated Higgs production as a tree-level probe of the Higgs–top-quark interaction [18, 20, 50–83]. Besides that, the $CP$ character of the top-Yukawa coupling can also be probed by investigation of the Higgs production via gluon fusion (ggF). While ggF production alone (without the association of jets) is only sensitive to the $CP$ nature of the top-Yukawa interaction via its total rate, ggF production in association with one[1] or two jets is more directly sensitive. In particular, Higgs production in association with two jets (ggF2j) is well known to be $CP$-sensitive via the difference of the azimuthal angles of the two jets that enables the construction of a $CP$-odd observable [56, 83, 85–97].

Generally, the ggF2j production does not directly probe the $CP$ structure of the top-Yukawa interaction but of the Higgs–gluon interaction. Conversely, a $CP$-violating Higgs–gluon interaction cannot only be induced by a $CP$-violating Higgs–top-quark interaction but also by $CP$-violating Higgs interactions with coloured BSM states. Yet, given the increasingly strong limits on the mass scale of coloured BSM states set by experimental searches at the LHC [98–103], any sign for $CP$ violation in the Higgs–gluon interaction is plausible to originate at least partially from the Higgs–top-quark interaction. This makes the ggF2j channel not only interesting in its own respect, but also in the context of constraining the $CP$ structure of the effective Higgs-gluon interaction and the top-Yukawa coupling.

Existing collider measurements so far only put relatively weak constraints on the $CP$ character of the Higgs–gluon interaction [38, 39, 46] (while EDM limits assuming the absence of $CP$ violation in all other Higgs couplings give competitive results [104]). In this work, we investigate how these limits could be improved. In particular, we compare the potential of two distinct kinematic regions (a VBF-like and a ggF2j-like kinematic region) and use well-established machine-learning techniques (boosted classifiers) to identify whether a given event originates from a $CP$-even interaction, a $CP$-odd interaction, or from their interference. This allows us to construct both $CP$-even and $CP$-odd observables. In the ggF2j-like region, these show a significantly better sensitivity than the difference of the azimuthal angles of the two jets, which is widely used in the literature. Moreover, we highlight how the separation into two kinematic regions helps to distinguish $CP$ violation in the Higgs–gluon interaction from $CP$ violation appearing in the Higgs interaction with massive vector bosons.

---

[1]For ggF production in association with one jet, jet substructure information has to be exploited to construct a $CP$-odd observable [84].

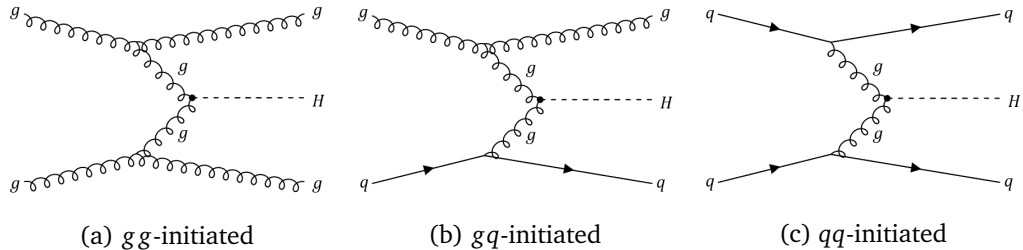

(a) $gg$-initiated  (b) $gq$-initiated  (c) $qq$-initiated

Figure 1: Examplary Feynman diagrams for the various ggF2j sub-channels.

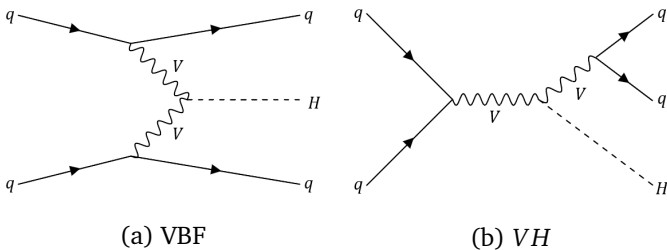

(a) VBF  (b) $VH$

Figure 2: Examplary Feynman diagrams for the considered background processes where $V \in [W^+, W^-, Z]$ and $q \in [q, \bar{q}]$.

The paper is structured as follows. In 2, we review Higgs production in association with two jets and introduce the effective theory that parameterizes our BSM couplings. Afterwards, we discuss the event generation and applied cuts in 3. 4 deals with the training setup of our classifiers. In this section, we also define the observables that are used to probe the $CP$-structure of the Higgs-gluon coupling. We present expected exclusion limits in 5. The interplay of the $CP$ structure of the Higgs–gluon interaction with the $CP$ structure of the Higgs couplings with massive vector bosons is discussed in 6. The interpretation of our limits in terms of limit on the Higgs–top-quark interaction and a comparison to current experimental limits is shown in 7. Finally, 8 concludes our findings and gives a brief outlook into possible future studies.

## 2 Gluon fusion with two jets in an EFT approach

Our work focuses on Higgs production with two jets. Here, we give an introduction to the relevant processes in the SM and then discuss our parameterization of BSM effects.

### 2.1 Gluon fusion in association with two jets

Higgs production via gluon fusion is the main production channel at the LHC with its cross section being (at minimum) an order of magnitude higher than any other Higgs production mode at $\sqrt{s} = 13$ TeV. Even if two or more jets associated with the parton level interaction are required in the final state, the cross section is still higher than all other Higgs production modes [105] ($\sigma_{\text{ggF}}^{j\geq2} = 7.88$ pb and $\sigma_{\text{VBF}} = 3.78$ pb at $\sqrt{s} = 13$ TeV). In the SM, the gluon fusion process is induced mainly by a top-quark loop, while for our work we consider an effective point-like interaction with unknown $CP$ state (see 2.2).

The ggF2j process can be classified by its initial state. Examplary diagrams for the three initial states can be seen in 1 where effective interactions are represented as a black dot. The individual initial states with gluons have higher cross sections due to the larger contribution of the gluons compared to those of the quarks to the parton distribution functions (PDFs)

of the proton. In our setup, we find for the SM case that the $gg$ initial state contributes $\sim 72\%$ to the total ggF2j cross section, while the $gq$ initial state still contributes $\sim 26.5\%$. The contribution of the $qq$ initial state is very small ($\sim 1.5\%$). We furthermore note that the interference between the $qq$-induced ggF2j production and Higgs production via VBF was found to be negligible [106, 107].

Example Feynman diagrams for the considered background (BG) processes — VBF and $VH$ production — are shown in 2.

## 2.2 Effective Higgs-vector boson interactions

We perform our study in an effective field theory (EFT) framework. Following [108], we parameterize the interaction of the Higgs boson with gluons in the following form,

$$\mathcal{L}_{ggH} = -\frac{1}{4v}\left(-\frac{\alpha_s}{3\pi}c_g G^a_{\mu\nu}G^{\mu\nu,a} + \frac{\alpha_s}{2\pi}\tilde{c}_g G^a_{\mu\nu}\tilde{G}^{\mu\nu,a}\right)H, \tag{1}$$

where $H$ is a Higgs field, $G^a_{\mu\nu}$ is the gluon field strength tensor, $v = 246$ GeV is the vacuum expectation value and $\alpha_s = g_s^2/4\pi$ is the strong coupling constant. Here, we assume that this interaction already includes the Higgs–gluon vertex induced by the top quark in the infinite top-mass limit.[2] Accordingly, $c_g = 1$ and $\tilde{c}_g = 0$ in the SM. A mixing angle $\alpha^{\text{Hgg}} = \tan^{-1}(\tilde{c}_g/c_g)$ can be defined to parameterize the $\mathcal{CP}$ state of the Higgs–gluon coupling. Deviations from the SM can either be induced by a modified top-Yukawa coupling or by coloured BSM particles.

The prefactors of the operators are chosen such that modifications of the top-Yukawa interaction directly map to the operators of 1 if we assume that no coloured BSM particles affect the Higgs–gluon interaction. In this case, we have $c_g = c_t$ and $\tilde{c}_g = \tilde{c}_t$ in the infinite top-quark mass limit, if we parameterize the top-Yukawa interaction via

$$\mathcal{L}_{\text{top-Yuk}} = -\frac{y_t^{\text{SM}}}{\sqrt{2}}\bar{t}(c_t + i\gamma_5\tilde{c}_t)tH, \tag{2}$$

where $y_t^{\text{SM}}$ is the SM top-Yukawa interaction and $c_t = 1$ as well as $\tilde{c}_t = 0$ for the SM.

We account for effects due to the finite mass of the top quark by rescaling the total rate and placing an upper bound on the Higgs transverse momentum (see 3).

The modified Higgs-gluon coupling of 1 affects ggF2j production. Its matrix element can be separated into three pieces

$$\left|\mathcal{M}_{\text{ggF2j}}\right|^2 = c_g^2\left|\mathcal{M}_{\text{even}}\right|^2 + 2c_g\tilde{c}_g\text{Re}\left[\mathcal{M}_{\text{even}}\mathcal{M}_{\text{odd}}^*\right] + \tilde{c}_g^2\left|\mathcal{M}_{\text{odd}}\right|^2. \tag{3}$$

The first and third terms proportional to the squared values of the coupling modifiers are $\mathcal{CP}$-even, while the second term parameterizing the interference between the two $\mathcal{CP}$-states is $\mathcal{CP}$-odd. $\mathcal{CP}$ violation in the Higgs-gluon interaction is therefore only realised for a non-zero value of the interference term. Correspondingly, the interference term gives a non-zero contribution only for distributions of $\mathcal{CP}$-odd observables.

---

[2]In other works (see e.g. [39, 46]), the Higgs–gluon interaction is written in the form $\mathcal{L}_{ggH} = -\frac{\alpha_s\pi}{v}\left(c_{gg}G^a_{\mu\nu}G^{\mu\nu,a} + \tilde{c}_{gg}G^a_{\mu\nu}\tilde{G}^{\mu\nu,a}\right)H$ without including the effect of the top-quark loop. In the infinite top-quark mass limit, both parameterizations are related via $c_g = 1 + 12\pi^2 c_{gg}$ and $\tilde{c}_g = -8\pi^2\tilde{c}_{gg}$.

In addition to a modification of the Higgs–gluon interaction, in 6 we will also consider a $\mathcal{CP}$-violating interaction of the Higgs boson with massive vector bosons. We parameterize this interaction as

$$\mathcal{L}_{\Phi VV} = \mathcal{L}_{\text{gauge}} + \frac{c_{\Phi\widetilde{W}}}{\Lambda^2}\mathcal{O}_{\Phi\widetilde{W}} \quad \text{with} \quad \mathcal{O}_{\Phi\widetilde{W}} \equiv \Phi^\dagger\Phi\widetilde{W}_{\mu\nu}W^{\mu\nu}, {}^{3} \tag{4}$$

where $\mathcal{L}_{\text{gauge}}$ is the SM gauge Lagrangian. $W$ and $\widetilde{W}$ are the $SU(2)_L$ field strength and its dual, respectively. $\Phi$ is the Higgs doublet and $\Lambda$ denotes the cut-off scale of the EFT, which we set to 1 TeV. If not stated otherwise, we will for the main part of this work assume that $c_{\Phi\tilde{W}} = 0$ such that the Higgs coupling to massive vector bosons is SM-like.

## 3 Event generation

As discussed in 1, we focus on Higgs production via gluon fusion in association with two jets, which we refer to as ggF2j in the following. This production channel offers a unique sensitivity to probe the $\mathcal{CP}$ nature of the Higgs–gluon interaction, since the two additional jets allow the construction of $\mathcal{CP}$-odd observables. Moreover, in this work, we concentrate on the Higgs decay to two photons. While this simplifies our analysis, our general analysis strategy can be straightforwardly adapted to other Higgs decay channels. As background processes, we consider Higgs production via vector-boson fusion (VBF) and Higgsstrahlung ($VH$). We leave a more in-depth study including the continuous background from the $qq \to qq$ process with two photons from final state radiation for future work.

We generate parton-level events at leading order and scale them to NLO by using flat K-factors from [56] with `MadGraph5_aMC@NLO` (version 3.4.0) [111]. The events are then passed on to `Pythia8` (version 8.306) [112] and `Delphes3` (version 3.4.2) [113] for parton showering, hadronisation and detector simulation. For the detector simulation with `Delphes3`, we employ the ATLAS detector card with a modified jet cone radius of $\Delta R = 0.4$ which is more widely used in experimental analyses. A jet is then defined as a set of hadronic particles in this cone which in total reach a minimum of $p_T^j \geq 20$ GeV. Reconstruction of the jet is performed using the anti-$k_t$ algorithm [114]. Photons (which are used for reconstructing the Higgs boson) are required to have a transverse momentum of $p_T \geq 0.5$ GeV.

Data sets are generated for Higgs production from the ggF2j, VBF and $VH$ processes. While the latter two do not receive any BSM contribution in our model, the event generation of ggF2j production is split into three terms following the parameterization in 3. We therefore obtain separate data sets for the terms proportional to $c_g^2$ (the $\mathcal{CP}$-even amplitude squared), $\tilde{c}_g^2$ (the $\mathcal{CP}$-odd amplitude squared), and $c_g\tilde{c}_g$ (the interference between the $\mathcal{CP}$-even and $\mathcal{CP}$-odd amplitude). This amounts to five distinct classes which we further split into independent data sets for training, validating and testing the classifiers, which will be introduced in 4. Further details on the event generation and the employed UFO model [115, 116] can be found in A.

Before training the classifiers, we impose a number of simple baseline cuts mimicking the event selection typically used by ATLAS and CMS for $H \to \gamma\gamma$ measurements [44, 45]. A cutflow table detailing the imposed cuts and the corresponding reduction in the event numbers can be found in 1, where the percentage of acceptance after each cut is displayed for all signal and background processes examined in this work.

First of all, a cut on $N_j$ and $N_\gamma$ ensures that at least two jets and two photons are found in each event, which are needed to reconstruct the Higgs boson and identify ggF2j production.

---

[3] At dimension six in the SM effective field theory (SMEFT), three distinct operators can introduce $\mathcal{CP}$ violation in VBF production (see e.g. [109]). In this work, we concentrate on the $\mathcal{O}_{\Phi\widetilde{W}}$ operator. We expect similar results for the other operators since all operators induce $\mathcal{CP}$-couplings operators with the same Lorentz structure (see e.g. [110] for more details).

Table 1: Cutflow table for the Higgs production mechanisms considered in this work. Listed are all cuts applied after the event generation, along with the percentage of events that survives the cut.

| | Fraction of accepted events | | | | |
|---|---|---|---|---|---|
| Applied cut | ggF2j $\|\mathcal{M}_{\text{even}}\|^2$ | ggF2j Interf. | ggF2j $\|\mathcal{M}_{\text{odd}}\|^2$ | VBF | $VH$ |
| Initial events | 100% | 100% | 100% | 100% | 100% |
| $N_j \geq 2; N_\gamma \geq 2$ | 48.1% | 50.8 % | 48.1% | 62.6% | 49.8% |
| 110 GeV $\leq m_{\gamma\gamma}$ $m_{\gamma\gamma} \leq 140$ GeV | 47.8% | 50.5% | 47.9% | 62.0% | 49.4% |
| $p_T^{\gamma 1}/m_{\gamma\gamma} \geq 0.35$ $p_T^{\gamma 2}/m_{\gamma\gamma} \geq 0.25$ | 39.4% | 40.9% | 39.8% | 50.0% | 40.5% |
| $p_T^{j1} \geq 30$ GeV $p_T^{j2} \geq 20$ GeV | 38.6% | 40.2% | 38.6% | 49.7% | 39.9% |
| $\|\eta_j\| \leq 2.5$ $\|\eta_\gamma\| \leq 2.5$ | 22.9% | 21.5% | 22.7% | 39.8% | 31.2% |
| $p_T^H \leq 200$ GeV | 18.6% | 18.4% | 18.3% | 34.4% | 26.8% |

Only accepting events with 110 GeV $\leq m_{\gamma\gamma} \leq 140$ GeV defines a region in which most Higgs events fall. The cuts on low values of $p_T^\gamma$ and $p_T^j$ suppress misidentified photons and jets that do not originate from the parton level interaction. Finally, only events with $|\eta_j| \leq 2.5$ and $|\eta_\gamma| \leq 2.5$ are accepted to match the pseudorapidity coverage of the ATLAS inner detector. As an additional requirement, we place an upper limit of $p_T^H \leq 200$ GeV on the Higgs transverse momentum (reconstructed out of the two photon momenta). This cut ensures that the top-quark loop inducing ggF2j production cannot be resolved and ggF2j production can be reliably interpreted using the effective Lagrangian given in 1 (see [117,118] for early calculations and e.g. the discussion in [119]).

As the cutflow in 1 shows, the cut on $p_T^H$ reduces the number of surviving events by $14 - 19\%$. While this reduction is not a critical limitation for our analysis, in principle one can include the events with higher $p_T^H$ beyond the infinite top-quark mass limit by employing the FT$_{\text{approx}}$ approximation [120]. This approximation combines the exact top-quark mass and width in the Born, one-loop and real amplitudes with approximated and re-scaled two-loop virtual contributions, originally applied to multi-Higgs production. The FT$_{\text{approx}}$ approximation has been successfully validated against the full NLO calculation in the SM for $H + j$ production [121]. Under the assumption that the FT$_{\text{approx}}$ approximation yields similarly accurate results for Higgs production in association with a higher multiplicity of jets, [121] has also improved the calculation of differential cross sections for $H + 2j$ by complementing the exact real corrections with the approximated two-loop virtual corrections. The good agreement between their NLO result in the heavy-top limit (HTL) and in the FT$_{\text{approx}}$ up to $p_T^H \leq 300$ GeV justifies our treatment in the HTL within the chosen cut on $p_T^H$. The relevance of the finite top-quark mass at higher $p_T^H$ and the success of FT$_{\text{approx}}$ in $H + j$ motivate a future study of $\mathcal{CP}$ properties of the Higgs boson without cutting out the high-$p_T^H$ events. For the scope of our work, our robust treatment with the $p_T^H$-cut results in conservative limits that might be improved by a refined event generation.

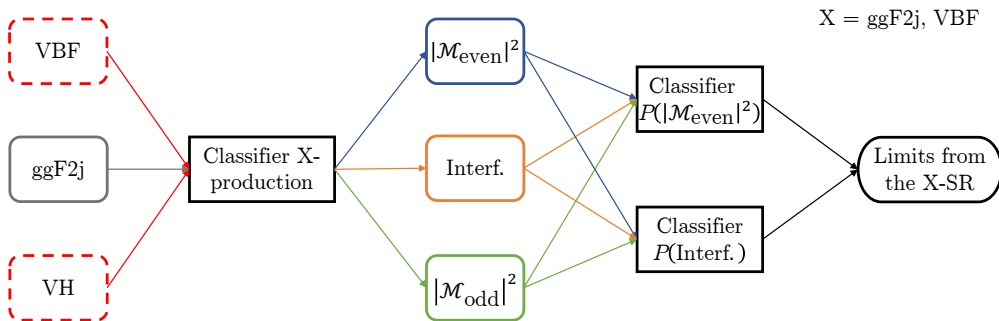

Figure 3: Outline of the strategy for this analysis. First, a classifier is trained for signal and background separation for each of the two considered Higgs production processes X = {ggF2j, VBF}. Afterwards, the obtained data in the respective kinematic region are used in two classifiers each, where one of them learns a $\mathcal{CP}$-even and the other one a $\mathcal{CP}$-odd observable.

## 4 Analysis strategy

Our analysis is structured into two steps. In the first step, two distinct classifiers are trained to separate the events originating from the different Higgs production channels. One classifier is trained to separate the signal events (from ggF2j production) from the considered Higgs background processes, which is dubbed the ggF2j signal region (SR). A second classifier is used to define a signal region which is enriched with $q\bar{q}$-induced ggF2j events. Although the ggF2j process is our signal process throughout the entire analysis, these events share the topology of VBF Higgs production, and we, therefore, train the classifier to identify VBF events. This is used to define what we call the "VBF-SR". Both SRs offer possible advantages in the analysis. In the ggF2j-SR, we expect an increased sensitivity as it contains a significant number of signal events. The VBF-SR is enriched with $q\bar{q}$ initial state events which were shown to carry the most $\mathcal{CP}$ information after applying typical VBF-cuts in an earlier work [91]. Events from the continuous background are expected to be enhanced in the ggF2j-SR compared to the VBF-SR, although we do not consider this background in this study.

Both classifiers are trained on the same input data. Since the two classifiers are independent, it is possible for events to appear in both or none of the kinematic regions. For our data set, the percentage of ggF2j data that appear in both the ggF2j-SR and the VBF-SR is about 8% of the total data set, while about 11% of the events do not appear in either of them. Subsequently, the events identified in each category are passed to two additional classifiers (see description below). These are used to distinguish the squared $\mathcal{CP}$-odd and $\mathcal{CP}$-even matrix elements and the interference term in the ggF2j production (see 3) and thereby to construct $\mathcal{CP}$-sensitive observables. The analysis steps for each of the kinematic regions are summarized in 3.

### 4.1 Signal-background separation

As mentioned above, two classifiers are trained to define a ggF2j- and a VBF-SR with distinct kinematics. In both cases, we reduce the separation to a binary classification problem, where the respective signal is trained against all other types of events. The classifiers are set up with

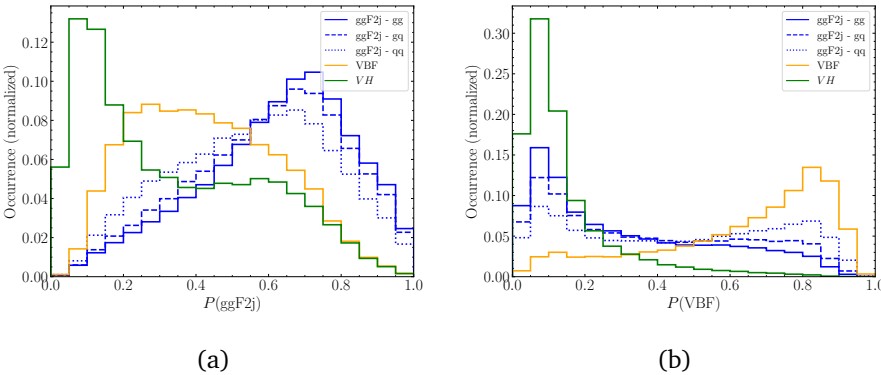

(a)             (b)

Figure 4: Scores of the trained classifiers for defining a (a) ggF2j and (b) VBF signal region for the different production channels.

`PyTorch` and the same data set is used for training, validation, and testing of both classifiers. The following kinematic variables are used as input for the signal-background separation:

- the energy $E$, the transverse momentum $p_T$, the pseudorapidity $\eta$, and the azimuthal angle $\phi$ of the Higgs boson and the two leading-$p_T$ jets,

- the invariant mass of the two leading-$p_T$ jets $m_{jj}$, their pseudorapidity difference $\Delta\eta_{jj}$, and their azimuthal angle difference $\Delta\phi_{jj}$ (where the sign is chosen such that the $\phi$ of the jet with the smaller $\eta$ is subtracted from the one with larger $\eta$),

- the number of jets in the event $N_j$, and

- the respective energy $E$ of each jet that is not leading or sub-leading in $p_T$.

Among the most important variables for the signal/background separation are $N_j$, $m_{jj}$, $|\Delta\eta_{jj}|$, and $\Delta\phi_{jj}$ (see e.g. [56]).[4] It can be seen that the three processes (ggF2j, VBF and $VH$) show quite distinct topologies which are important for training the NN. While the usage of higher-level observables (like $|\Delta\eta_{jj}|$ and $m_{jj}$) does not provide the classifiers with new information, they can still be useful in order to improve the training process. Such observables should, however, only be included if they are expected or known to impact the outcome of the classifier, as they can slow down the training otherwise. In the present case, $|\Delta\eta_{jj}|$ and $m_{jj}$ are well-known examples for variables allowing one to distinguish the events stemming from VBF and ggF2j production.

The classifier creating the ggF2j-SR (VBF-SR) reaches an accuracy of about 70% (79%). The classifier score, which gives an estimate of the probability for each event to be a signal event, is calculated for ggF2j, VBF and $VH$ Higgs production, respectively, and plotted in 4a (4b). The ggF2j production is additionally split up into the three possible initial states ($qq$, $gq$ and $gg$). We observe that the more quarks are in the initial state, the more likely it is for an event to be classified as VBF. Especially for the $qq$ initial state, this can be easily understood by comparing the example Feynman diagrams in 1c and 2a, which only differ via the exchange of the gluon/heavy vector boson propagators creating the Higgs. We additionally observe that interference events from ggF2j are more likely to be identified as VBF-like events than events from the squared terms are.

For both classifiers, the respective signal process is the dominant process for a score of $P(\text{signal}) \gtrsim 0.5$ and we therefore set $P(\text{signal}) \geq 0.5$ as a cut to define our SRs. All accepted events are combined into a new data set and subsequently passed on to two additional classifiers, in order to gain information about their $\mathcal{CP}$ structure.

---

[4]For more advanced approaches using e.g. jet energy profiles or jet charge see [122, 123].

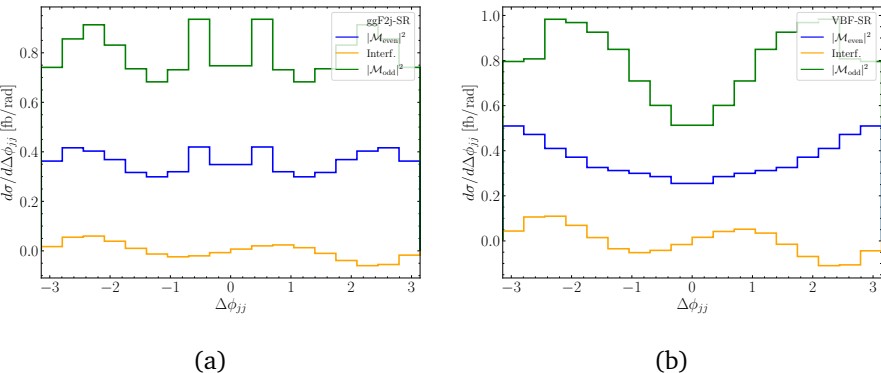

(a)          (b)

Figure 5: The differential cross section as a function of $\Delta\phi_{jj}$ in the (a) ggF2j- and (b) VBF-SR, plotted for the $|\mathcal{M}_{\text{even}}|^2$ (blue), interference (orange) and $|\mathcal{M}_{\text{odd}}|^2$ (green) contributions.

## 4.2  Separating the different ggF2j contributions

The three terms in the squared ggF2j amplitude in 3 can be probed by exploiting rate information as well as their different kinematics. The interference term can only be probed by $\mathcal{CP}$-odd observables since its positive and negative contributions cancel out for $\mathcal{CP}$-even observables. One such observable is $\Delta\phi_{jj}$, which has been used as a $\mathcal{CP}$ observable in previous analyses (see e.g. [38]). 5 shows the contributions of the squared and interference terms to the differential cross section, plotted against $\Delta\phi_{jj}$. Since $\Delta\phi_{jj}$ is a $\mathcal{CP}$-odd observable, the distribution for the interference term is antisymmetric, while the distribution of the squared $\mathcal{CP}$-even and $\mathcal{CP}$-odd matrix elements are symmetric. We impose a $\mathcal{CP}$ flip on the events[5] before the training to ensure the symmetry of the distributions even for a limited number of Monte-Carlo events. The two peaks close to $\Delta\phi_{jj} = 0$ in the ggF2j-SR originate from events with low $m_{jj}$ and vanish if a minimal $m_{jj}$ cut is imposed.

For the training of the $\mathcal{CP}$ classifiers, we use the full kinematic information of the two leading jets and the reconstructed Higgs boson, as well as $m_{jj}$, $\Delta\eta_{jj}$, and $\Delta\phi_{jj}$ as additional high-level observables. The distributions of all variables used in the training can be found in B where they are split up into the three possible contributions to the ggF2j cross section. Compared to the signal-background separation, we dropped the information about additional jets, since these were found to carry little to no information about the $\mathcal{CP}$ character of the Higgs–gluon interaction. For the $\mathcal{CP}$ classification, we used the `Gradient Boosting Classifier` from the `scikit-learn` [124] package. We train two classifiers independently for each of the signal regions defined by the signal-background classifiers. Two observables are constructed from their output in the following way:

- The first classifier only separates between the squared $\mathcal{CP}$-even and the squared $\mathcal{CP}$-odd terms. The corresponding output observable is defined as $P(c_g^2)$, which can be interpreted as a probability that a given event originates from the $|\mathcal{M}_{\text{even}}|^2$ contribution.

- The second classifier deals with the interference term. It is trained to separate positive and negative interference events as well as to distinguish these from events originating from either of the squared terms. Following [125], the observable built from the output is defined as $P_+ - P_-$ where $P_+$ ($P_-$) is the probability of an event to correspond to positive (negative) interference. This observable is $\mathcal{CP}$-odd by definition.

---

[5]This corresponds to flipping the sign of the particle momenta as well as in case of the interference term the weight on an event-by-event basis.

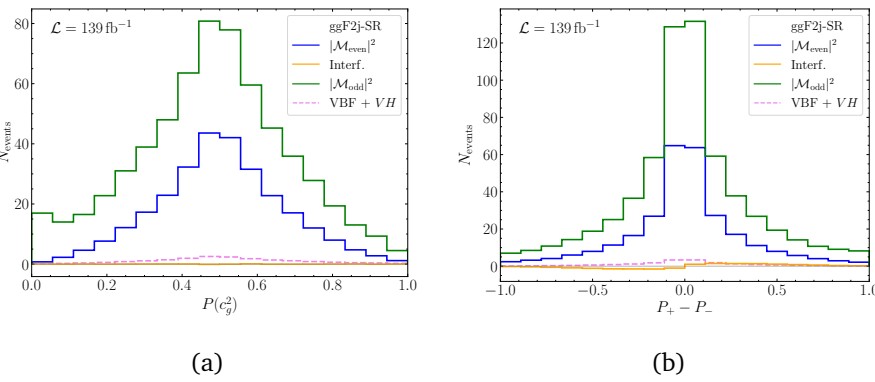

Figure 6: Distributions of the two $\mathcal{CP}$ discriminants (see text for the definition) for the classifiers yielding the strongest limits on the coupling modifiers in the ggF2j-SR.

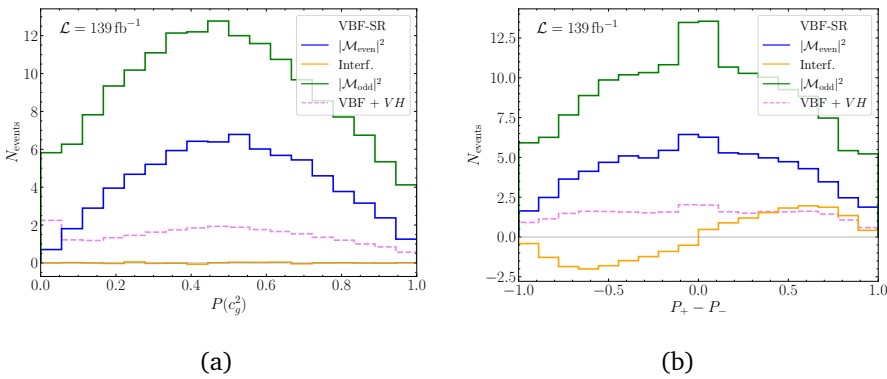

Figure 7: Distributions of the two $\mathcal{CP}$ discriminants for the classifiers giving the strongest limits on the coupling modifiers in the VBF-SR.

A similar approach has been followed in [126,127], which also connect it to the matrix element approach.

6 shows the output of the two $\mathcal{CP}$ classifiers in the ggF2j-SR. Each classifier has been trained 100 times (see C) and here we show the one with the strongest constraints on the Higgs-gluon coupling modifiers (see 5.1). For the $P(c_g^2)$ classifier (see left panel of 6), this corresponds to the case in which the difference between the squared terms was learned best, while the contribution from the interference term is zero. When applying the same normalization to both of the squared terms, the distribution from the $\tilde{c}_g^2$ term shows to be higher towards low values of $P(c_g^2)$ and lower around $P(c_g^2) = 0.5$. The difference between the classifier with the strongest constraints and other trained classifiers is mainly in the shape of the left outermost bin.

The $P_+ - P_-$ classifier (see right panel of 6) learning the interference terms showed only very slight fluctuations during the training processes. Since its output is a $\mathcal{CP}$-odd observable by construction, the squared ggF2j contributions, as well as BG contributions, show the expected behaviour of being symmetric around zero. The interference contribution is asymmetric around zero. Its amplitude compared to the squared contributions is very small due to its small cross section, an overall lower number of interference events in the ggF2j-SR from the signal-background separation, and misidentified events during the $\mathcal{CP}$ classification.

The output of the $\mathcal{CP}$ classifiers in the VBF-SR is plotted in 7. Again, we show the distributions of the classifiers yielding the strongest limits (see 5.2) after 100 training iterations. The shapes of the two squared distributions show similar differences as in the ggF2j-SR with the

distribution stemming from the $\tilde{c}_g^2$ contribution being higher towards low values of $P(c_g^2)$ and lower around and above $P(c_g^2) = 0.5$. Here, the visible statistical fluctuations in the $|\mathcal{M}_{\text{even}}|^2$ and $|\mathcal{M}_{\text{odd}}|^2$ contributions arise due to the comparably low number of ggF2j events in this signal region.

In contrast, the asymmetry of the interference term in the $P_+ - P_-$ observable is much more visible. Even in the VBF-SR, the number of BG events (consisting of VBF and VH events) is still lower than the number of ggF2j events, which is a consequence of the smaller cross section.

## 5 Sensitivity to the Higgs–gluon coupling

In this section, we evaluate the sensitivity to the Higgs–gluon couplings. Assuming SM data, we fit the coupling modifiers to the expected measurements of the two signal regions as well as their combination resulting in expected limits. Details on the parameter fit can be found in D. Furthermore, we evaluate the impact of individual observables on the expected limits.

### 5.1 ggF2j signal region

We first present our results in the ggF2j-SR, where the contributions from background processes are strongly suppressed by the signal-background classifier. Expected limits are derived under the assumption that the data is SM-like for a luminosity of 139 fb$^{-1}$. This fit is based on the binned distribution of the various observables. The results are then plotted in the $(c_g, \tilde{c}_g)$ parameter plane.

First, we show the limits obtained from the $\Delta\phi_{jj}$ distribution (see 5a) in 8a. It can be seen that the allowed parameter space is constrained to the form of an ellipse for which the ggF2j total rate is close to its SM value. The $\Delta\phi_{jj}$ observable alone is not able to exclude any region of this ellipse within the $1\sigma$ region.

The limits from the two classifiers are obtained from 6a and 6b, respectively. The $P(c_g^2)$ observable provides by far the strongest constraints among the two observables constructed from the classifiers (see 8b). Here, the ellipse is split up within the $3\sigma$ region, which constrains the $\mathcal{CP}$-odd Higgs–gluon coupling $\tilde{c}_g$ to the interval $[-0.35, 0.35]$ at the $1\sigma$ level. While the $P_+ - P_-$ observable based on the interference contribution leads to weaker constraints (see 8c), it still outperforms the $\Delta\phi_{jj}$ variable. Computing the limits from a two-dimensional histogram with both $\mathcal{CP}$-observables shows a small improvement over the $P(c_g^2)$ limit (see 8d), limiting $\tilde{c}_g \in [-0.32, 0.32]$ at the $1\sigma$ level.

### 5.2 VBF signal region

Since the VBF-SR contains significantly more VBF events than the ggF2j-SR, the difference in the rate between the BSM and SM scenarios, which both contain the same VBF and $VH$ events, is reduced compared to the ggF2j-SR. This is reflected in 9 where the ellipse is now wider.

Just like in the ggF2j-SR, the $\Delta\phi_{jj}$ variable in 9a is not able to exclude part of the ellipse. In contrast to the previous results, the $P(c_g^2)$ observable gives very similar limits to $\Delta\phi_{jj}$ (see 9b). The interference classifier (see 9c) again results in worse limits than the $P(c_g^2)$ observable, but now even worse limits than $\Delta\phi_{jj}$. This can be understood by taking into account which information each of the observables is provided with. While the classifiers directly use $\Delta\phi_{jj}$ during the training process, the classifier training $|\mathcal{M}_{\text{even}}|^2$ vs. $|\mathcal{M}_{\text{odd}}|^2$ is missing the information about the interference contribution. This effect is negligible in the ggF2j-SR due to the small amplitude of the interference term, but it is non-negligible here (see 6b and 7b). On the other hand, the interference classifier does not differentiate between the $|\mathcal{M}_{\text{even}}|^2$

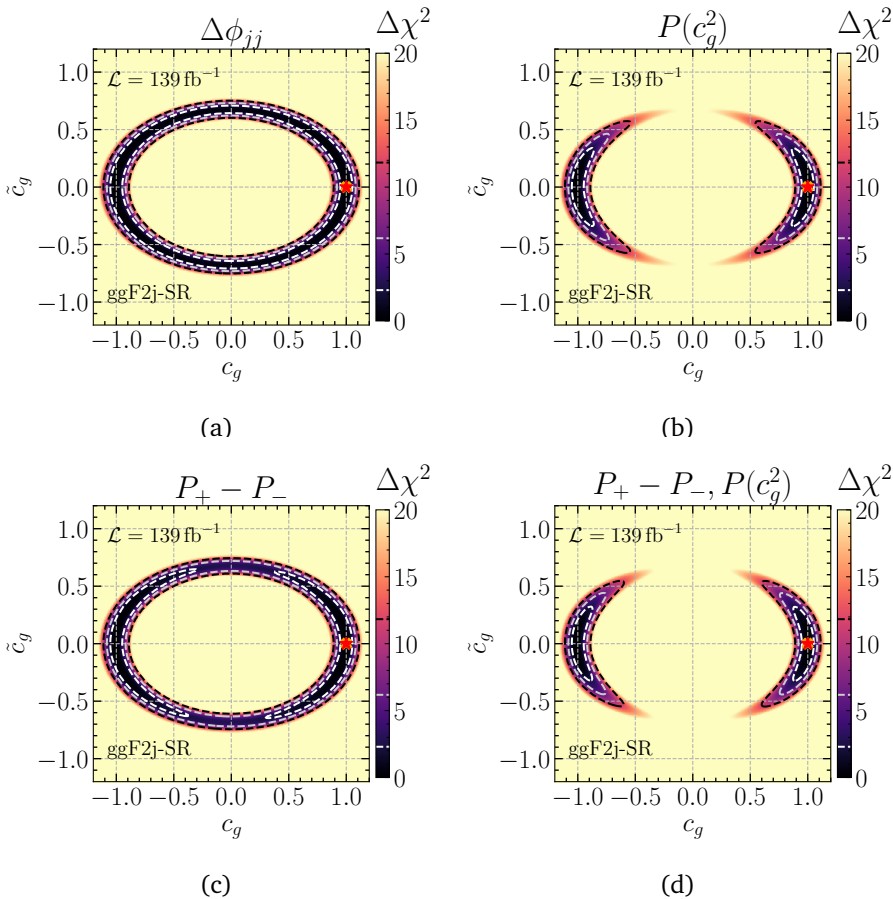

Figure 8: Limits from (a) $\Delta\phi_{jj}$, (b) the classifier trained to distinguish $\left|\mathcal{M}_{\text{even}}\right|^2$ vs. $\left|\mathcal{M}_{\text{odd}}\right|^2$, (c) the classifier trained to distinguish positive vs. negative interference, and (d) the combined limits from both classifiers. All limits are shown for the ggF2j-SR. The 1-, 2- and 3-$\sigma$ contours are marked by white, grey and black dashed lines, respectively. The SM is marked by an orange cross and the best-fit (BF) point by a red star.

and $\left|\mathcal{M}_{\text{odd}}\right|^2$ contributions. Finally, the combination of both classifiers (see 9d), which uses the full information about the contributing terms to $\sigma_{\text{ggF2j}}$, puts stronger constraints on the Higgs–gluon coupling modifiers than $\Delta\phi_{jj}$. In contrast to the ggF2j-SR, the improvement over $\Delta\phi_{jj}$ is, however, not so significant. This highlights the high sensitivity of the $\Delta\phi_{jj}$ variable in the VBF-SR (see also the discussion in 5.4). We obtain $\tilde{c}_g \in [-0.58, 0.58]$ for the best classifier in this signal region.

## 5.3 Combination

We now evaluate how the constraints on the Higgs–gluon coupling modifiers change when we combine the ggF2j-SR and the VBF-SR. The limits obtained from the $P(c_g^2)$ and $P_+ - P_-$ classifiers can be found in 10a and 10b, respectively. The limits combining the $P(c_g^2)$ and $P_+ - P_-$ classifiers are depicted in 10c. The $1\sigma$ ($2\sigma$) limit is shown as a dashed (dotted) line for the ggF2j-, the VBF- and the combined SRs.[6] As the plots display, the combination of both SRs

---

[6]In the two left panels, the constraints from the ggF2j-SR and the combined SRs overlap such that the two curves look like a solid black line.

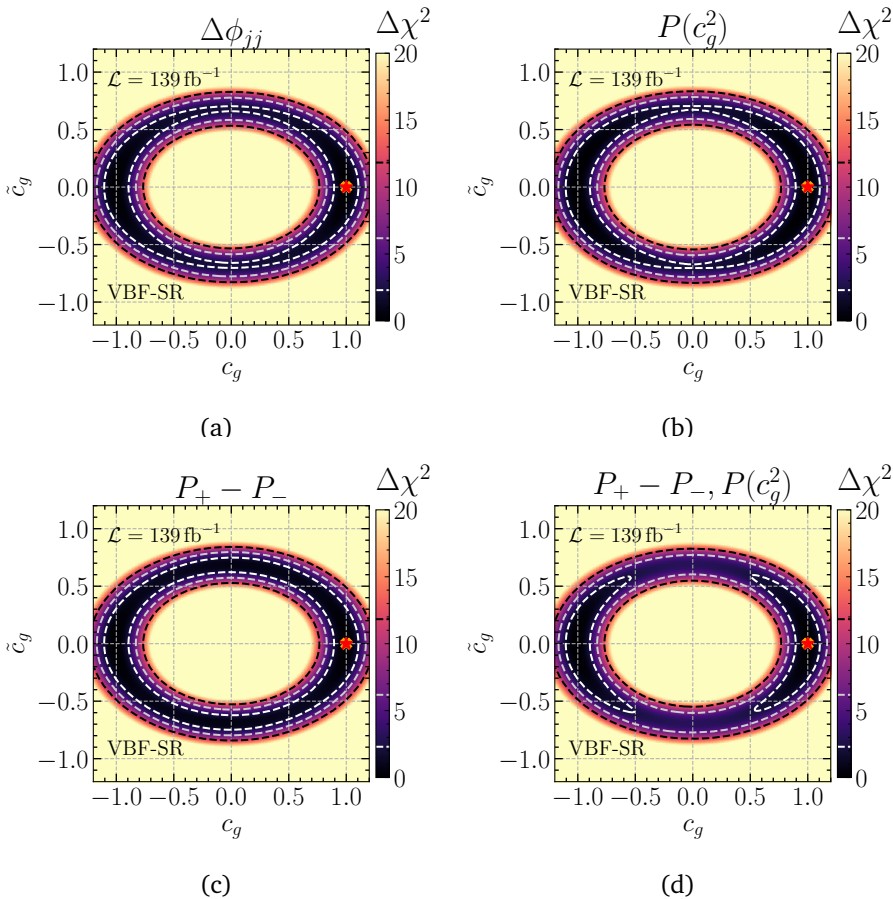

Figure 9: Limits from (a) $\Delta\phi_{jj}$, (b) the classifier trained to distinguish $\left|\mathcal{M}_{\text{even}}\right|^2$ vs. $\left|\mathcal{M}_{\text{odd}}\right|^2$, (c) the classifier trained to distinguish positive vs. negative interference and (d) the combined limits from both classifiers. All limits are shown for the VBF-SR. The 1-, 2- and 3-$\sigma$ contours are marked by white, grey and black dashed lines, respectively. The SM is marked by an orange cross and the BF point by a red star.

leads to an advantage over the single ggF2j-SR only for the interference classifier. In contrast to this, the limits from the $P(c_g^2)$ observable in the combined region are slightly weaker than in the ggF2j-SR alone. These opposite effects of the combined SR on the two classifiers originate from a different fraction of the ggF2j, VBF and $VH$ events when adding the events from the VBF-SR to the ggF2j-SR: The relative number of both VBF events and ggF2j interference events is increased, as the latter make up the majority of misidentified ggF2j events in the VBF-SR. The limits obtained from using both classifier observables are not affected when combining both signal regions, see 10c.

So far we have only looked at exclusion limits obtained for the scenario that the measured data is SM-like. As an addition, we show in 10d the limits of the combined classifier in the three signal regions, for the case that $\alpha^{Hgg} = 45°$ is realised in Nature. While, as before, the constraints from the VBF-SR are a full ellipse and therefore allow the SM point at both the $1\sigma$ and $2\sigma$ levels, the ggF2j-SR excludes the SM at the $1\sigma$ level. The combined SR is able to exclude the SM at the $2\sigma$ level.

Going back to the case of SM data, the expected limits at different benchmark luminosities corresponding to Run-2 and Run-3 of the LHC and further the HL-LHC are depicted in 11. We note that the improvement of the limits with increased luminosity is slightly worse than naively

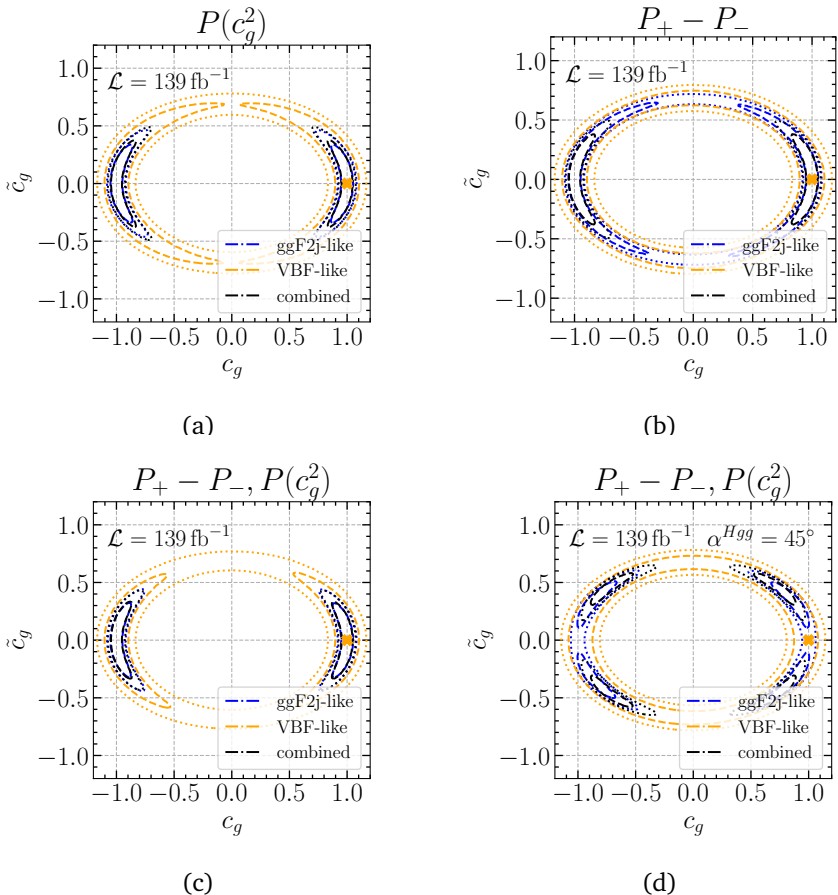

Figure 10: Comparison of the limits on the coupling modifiers in the ggF2j-, VBF- and combined SRs for the $P(c_g^2)$ (a) and $P_+ - P_-$ (b) observables, as well as their combination (c) for SM data and data generated for $\alpha^{Hgg} = 45°$ (d). The $1\sigma$ regions are shown as dashed lines, while the dotted lines correspond to the $2\sigma$ confidence levels. The SM is marked by an orange cross.

expected ($\propto 1/\sqrt{\mathcal{L}}$). This is a consequence of the suppression of the $|\mathcal{M}_{\text{odd}}|^2$ contribution close to the SM point as well as the small impact of the interference term. We expect the limits at the $2\sigma$ level to improve from $\tilde{c}_g \in [-0.44, 0.44]$ at $\mathcal{L} = 139$ fb$^{-1}$ to $\tilde{c}_g \in [-0.35, 0.35]$ at $\mathcal{L} = 300$ fb$^{-1}$ and $\tilde{c}_g \in [-0.2, 0.2]$ at $\mathcal{L} = 3000$ fb$^{-1}$, respectively. It should be noted, however, that the limits derived here are expected to degrade once the non-Higgs background and systematic uncertainties are taken into account. We leave a full investigation of these effects for further work.

## 5.4 Highest-impact observables

Here, we evaluate the impact of the various observables used by the classifiers on the output of the $\mathcal{CP}$ classifier. We evaluate this impact with the SHAP package [128], which explains the output in terms of Shapley values from cooperative game theory [129, 130]: Each classifier has a fixed "worth" determined from its separation power when all variables are used in the training. The Shapley value of one variable $x_i$ is then determined by taking the sum over all possible subsets of the parameter space not containing $x_i$ and evaluating the loss in "worth" compared to the full case. Taking every possible subspace into account guarantees that possible correlations between variables do not falsify the result. For more details, we refer to [131].

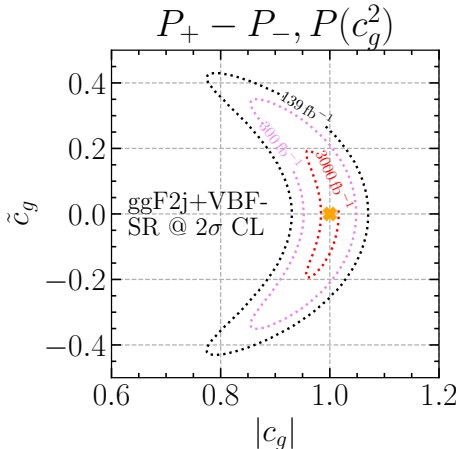

Figure 11: Comparison of the limits on the coupling modifiers in the combined SR for projections to higher luminosities. The dotted lines correspond to the $2\sigma$ confidence levels. The SM is marked by an orange cross.

In the following, the Shapley values are referred to as SHAP values to meet the nomenclature of the SHAP package.

The SHAP plots are structured as follows:

- Individual variables are plotted against their SHAP value. A point at a high absolute SHAP value means that the variable had a large impact on deciding what label the event has been given by the classifier. The sign of the SHAP value determines which label was chosen, i.e., for the $|\mathcal{M}_{\text{even}}|^2$ vs. $|\mathcal{M}_{\text{odd}}|^2$ classifier, a positive sign signals that $|\mathcal{M}_{\text{even}}|^2$ is more likely to be chosen.

- The color of the points represents the value of the variable itself. Red points stand for high values of the variable, while blue ones stand for low ones.

- The variables are ranked by the mean of the absolute SHAP value. Therefore, variables which have many events (displayed as bulks) deviating from zero are identified as the most important ones. Single points of variables that reach high absolute SHAP values are understood as outliers and only have a minor impact.

We first focus on the classifier separating the squared ggF2j terms in the respective signal regions in 12. In the ggF2j-SR (see 12a) the most important variables are $p_T$ and $E$ of the leading-$p_T$ jets, as well as their invariant mass, while the observables associated with the Higgs boson play a subordinate role. This picture is flipped in the VBF-SR (see 12b) where now $p_T$ and $E$ of the Higgs boson are the most important variables, followed by the jet kinematics. Note that the overall absolute SHAP values are relatively low compared to their spread indicating that no single observable drives the outcome of the classifiers. Instead, the full kinematic information is needed to separate between the squared ggF2j terms.

In the case of the interference classifiers (see 13), $\Delta\phi_{jj}$ is by far the most important variable for both signal regions. As a $\mathcal{CP}$-odd observable by construction, $\Delta\phi_{jj}$ is expected to give a good separation between the positive and negative parts of the interference. In the ggF2j-SR (13a), other important variables are the $p_T$ of the Higgs boson, as well as the pseudorapidity of the jets — in the form of the combination $\Delta\eta_{jj}$ as well as $\eta^{j_1}$ and $\eta^{j_2}$ alone. For the VBF-SR (13b), it should be noted that the mean of the $\Delta\phi_{jj}$ distribution peaks at much higher SHAP values than in the ggF2j-SR. This is in agreement with the limits obtained in 5.2, where the

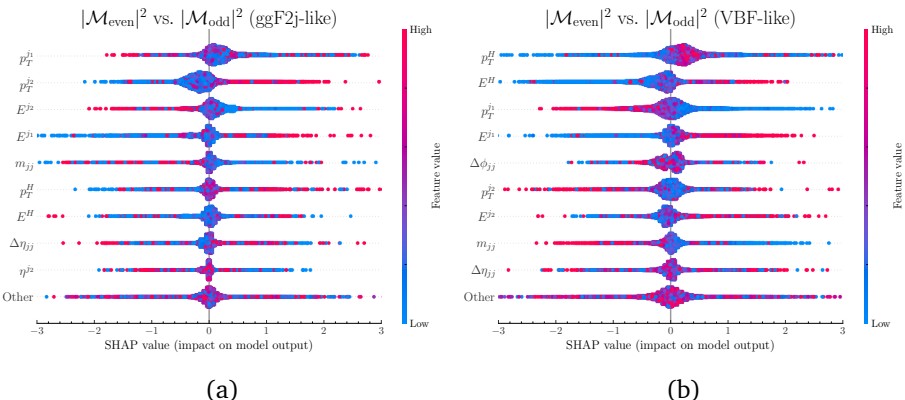

Figure 12: SHAP values for the variables used in classifiers separating the $|\mathcal{M}_{\text{even}}|^2$ and $|\mathcal{M}_{\text{odd}}|^2$ terms. The classifiers are trained in (a) the ggF2j-SR and (b) the VBF-SR. High absolute SHAP values indicate a strong influence on the classifier score. The color of individual points marks the value of the specific variable. The variables are ordered by taking the mean of the absolute SHAP value. All variables used in the training which are not shown in the plot are summed up and marked as "Others".

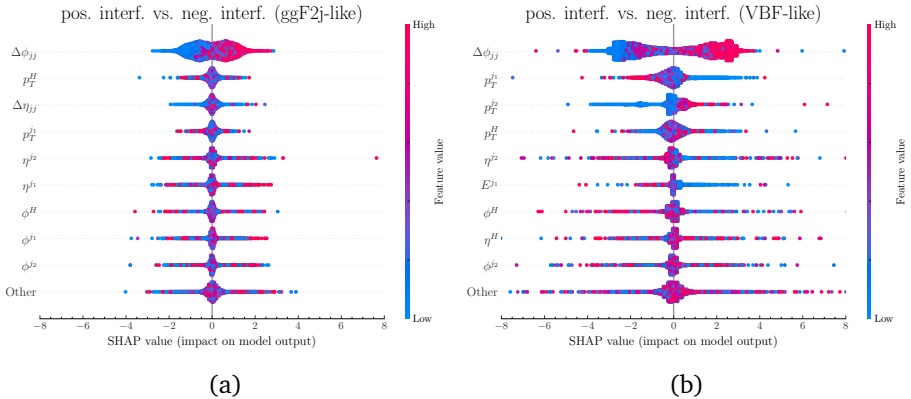

Figure 13: Same as 12, but the SHAP values for the classifiers separating the positive and negative interference terms are shown.

limits from the classifier using the full event kinematics showed only a slight improvement over the limits from $\Delta\phi_{jj}$ alone. The next two most sensitive variables are $p_T^{j_1}$ and $p_T^{j_2}$. Our findings agree with [132], in which $\mathcal{CP}$-violating effects in the Higgs coupling to $W$ bosons were studied with a different machine learning approach, identifying $\Delta\phi_{jj}$, $p_T^{j_1}$ and $p_T^{j_2}$ as the three most sensitive observables for this coupling when using the product of all three variables.

# 6 Disentangling $\mathcal{CP}$ violation in the $ggH$ and $HVV$ couplings

So far, we have concentrated on $\mathcal{CP}$ violation in the Higgs–gluon interaction. However, BSM physics may affect multiple Higgs couplings at once. As discussed above, VBF and $VH$ production are the main Higgs background processes for investigations of ggF2j production. While, as mentioned in 1, a lot of effort has been put forward already to investigate the $\mathcal{CP}$ structure of the Higgs couplings to massive vector bosons, we investigate in this section how the presence of $\mathcal{CP}$ violation in the $HVV$ couplings influence our limits on the Higgs–gluon coupling in the two signal regions.

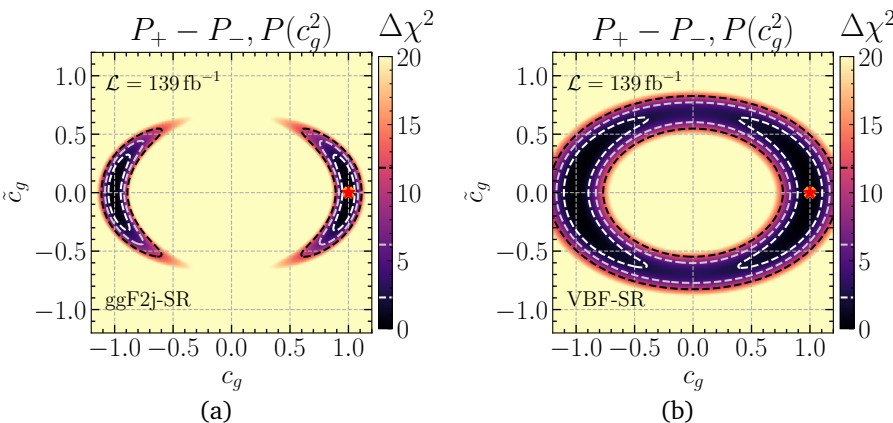

Figure 14: Combined limits of the two classifiers for the case that $\mathcal{CP}$ violation is present in the $HVV$ coupling in the (a) ggF2j-SR and the (b) VBF-SR.

For this, we generate additional data sets for VBF and $VH$ production in the context of the SMEFT, where a non-zero value of the $c_{\Phi\tilde{W}}$ Wilson coefficient introduces $\mathcal{CP}$ violation in the Higgs coupling to $W$ bosons (for details see A). These data sets are added to the SM-like VBF and $VH$ data sets which have been used in our analysis so far. To quantify the effects on our Higgs–gluon coupling limits, we do not re-train our classifiers, but directly build new observables based on the adjusted events.

14 depicts the combined limits of the two classifiers obtained in the ggF2j-SR and the VBF-SR when using the data sets with $\mathcal{CP}$ violation in the $HVV$ coupling. We obtain our limits in the $(c_g, \tilde{c}_g)$ parameter plane by keeping $c_{\Phi\tilde{W}} \in [-1, 1]$ as a free parameter in the fit and then profiling over it. In the ggF2j-SR, we observe no change in the limits compared to the previous results (compare 14a with 8d). Although the classifiers are not retrained with the $\mathcal{CP}$ violating $HVV$ samples, we find only a low number of events from those samples in the ggF2j-SR. Varying $c_{\Phi\tilde{W}} \in [-1, 1]$[7] is therefore not enough to induce a significant change in the limits.

The limits based on the VBF-SR with $\mathcal{CP}$ violation in the $HVV$ coupling are plotted in 14b. In the VBF-SR, the amount of $\mathcal{CP}$ violating $HVV$ events is enriched even without explicitly training the classifier to recognize them. For the $P(c_g^2)$ classifier, no changes in the limits can be seen since the interference contributions cancel in a $\mathcal{CP}$-even observable. We do observe changes in the $\Delta\chi^2$ of the $P_+ - P_-$ classifier, but these are hard to see by eye due to the closed ellipse in 9c. We therefore only show the combined limits here, in which a clear difference can be observed between the $1\sigma$ regions of 9d and 14b. Specifically, the $1\sigma$ limits are weakened from $\tilde{c}_g \in [-0.58, 0.58]$ to $\tilde{c}_g \in [-0.64, 0.64]$.

We conclude that the limits in the $(c_g, \tilde{c}_g)$ parameter plane get slightly weaker only in the VBF-SR if a Wilson coefficient parameterizing $\mathcal{CP}$ violation in the $HVV$ coupling is kept free-floating in the fit. This is expected since VBF-like events are strongly suppressed in the ggF2j-SR, which is, therefore, robust against possible $\mathcal{CP}$ violation in the $HVV$ coupling.

# 7 Limits on the $\mathcal{CP}$ structure of the top-Yukawa coupling

As discussed in 2.2, we normalized the coefficients of the effective Higgs–gluon interaction (see 1) such that they directly correspond to the modifiers of the top-Yukawa coupling (see 2) if no low-mass coloured BSM particles are present and if we neglect the contributions of the

---

[7]Current experimental limits are well within this range [41, 133].

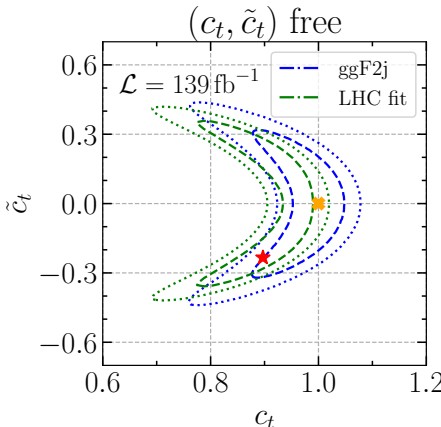

Figure 15: Comparison of the projected combined ggF2j limits (blue) to the current global limits from LHC data (green). Shown are the $1\sigma$ (dashed) and $2\sigma$ regions (dotted) in both cases. The SM is marked by an orange cross and the BF point of the LHC data by a red star.

lighter quarks. Since experimental limits on coloured BSM particles are becoming increasingly strong [98–103], these particles can also only induce a small $\mathcal{CP}$-odd Higgs–gluon coupling. Taking as an example a coloured fermion with a mass of 1 TeV and a Yukawa-type coupling of $\mathcal{O}(1)$ to the Higgs boson, we expect this fermion to contribute to the Higgs–gluon couplings at the $\mathcal{O}(v^2/\Lambda^2) \sim 0.1$.

Assuming that the contribution of coloured BSM particles is negligible, we can reinterpret our projected limits on the Higgs–gluon coupling as projected limits on the top-Yukawa coupling (by simply replacing $c_g$ by $c_t$ and $\tilde{c}_g$ by $\tilde{c}_t$). Then, our projected limits on the $\mathcal{CP}$ mixing angle $\alpha^{\text{Htt}} = \tan^{-1}(\tilde{c}_t/c_t)$ for a luminosity of 139 fb$^{-1}$ result in

$$\alpha^{\text{Htt}}_{\text{ggF2j}} \in [-15°, 15°], \quad \alpha^{\text{Htt}}_{\text{VBF}} \in [-25°, 25°] \ @ \ 68\% \ \text{CL}, \tag{5}$$

for the respective signal regions.

Besides exploiting the kinematics of ggF2j production, also the total rate information of Higgs production via gluon fusion (without jets) can be used to constraint the $\mathcal{CP}$ structure of the top-Yukawa coupling (see e.g. the recent [20, 75]). These global fits — besides other measurements — take into account total rate information from Higgs production via gluon fusion and the Higgs decay to two photons, which are very sensitive, but model-dependent constraints.

To provide an up-to-date comparison to our ggF2j results, we have updated the fits performed in [20, 75] using the new `HiggsSignals` which is part of `HiggsTools` and contains additional recent experimental results [134–136].[8] The result is shown in the $(c_t, \tilde{c}_t)$ parameter plane in 15. The blue contours depict the limits from our ggF2j analysis, and the green ones are based on the global fit to existing experimental results. As discussed in detail in [75], the fit constraints are dominated by total rate measurements of Higgs production via gluon fusion and the Higgs decay to two photons.[9] 15 shows that a ggF2j $\mathcal{CP}$ measurement has the potential to provide sensitivity to a $\mathcal{CP}$-odd top-Yukawa coupling that is competitive with a more model-dependent global fit to Higgs cross-section and branching ratio measurements.

---

[8]We are using version 1.1 of the `HiggsSignals` data set.

[9]The observed shift of the $1\sigma$ and $2\sigma$ regions to lower values of $c_t$ is caused by recent experimental measurements of top-associated Higgs production and Higgs production via gluon fusion with a lower rate than expected in the SM [42, 137].

# 8 Conclusions

The $\mathcal{CP}$ nature of the Higgs boson is among the few Higgs properties which are still only loosely constrained. One coupling, which is especially important for LHC physics, is the Higgs–gluon interaction — facilitated at the loop level. The $\mathcal{CP}$ structure of this interaction can be probed via Higgs production in association with two jets (ggF2j).

In this work, we studied how the information about the $\mathcal{CP}$ structure of the Higgs–gluon interaction can be best extracted. Focusing on the Higgs decay to two photons, we first trained a classifier to define a ggF2j-enriched signal region differentiating it from VBF and $VH$ production as the relevant backgrounds. In order to fully exploit the quark-initiated ggF2j channel, which is kinematically very similar to VBF production, we also defined a VBF-enriched signal region.

For both SRs, we trained two classifiers to separate the different contributions to the squared ggF2j amplitude: the square of the $\mathcal{CP}$-even amplitude, the square of the $\mathcal{CP}$-odd amplitude, and the interference contribution. Out of these classifiers, we constructed one $\mathcal{CP}$-even and one $\mathcal{CP}$-odd observable. Based on the distributions for these observables, we derived expected upper limits. These constrain the $\mathcal{CP}$-odd Higgs–gluon coupling modifier to $|\tilde{c}_g| \leq \{0.35, 0.28, 0.15\}$ at the $2\sigma$ level for integrated luminosities of $\{139 \, \text{fb}^{-1}, 300 \, \text{fb}^{-1}, 3000 \, \text{fb}^{-1}\}$, respectively. We found these to significantly outperform limits based only on the difference in the azimuthal angle of the two leading jets, which is an observable commonly employed in the literature. Moreover, the ggF2j-SR has a significantly higher sensitivity than the VBF-SR, which is expected by the higher number of ggF2j events. These results suggest that improvements in current experimental limits are possible with well-established techniques. The extent of these improvements will depend on factors such as the realistic treatment of non-Higgs backgrounds and systematic uncertainties, which are left for future work.

We note that our limits could be further improved by including other Higgs decay channels and removing the cut on the Higgs transverse momentum. Moreover, advanced analysis techniques like e.g. the matrix-element approach [63, 138–146] or machine-learning-based inference [147–151] could be used. We leave such improvements for future work.

In addition, we used interpretable machine learning (in the form of SHAP values) to investigate which observables have the highest impact on the classifications. While the azimuthal angle of the two leading jets is by far the most important observable for separating the interference term from the squared terms, also other momenta like the transverse momenta of the jets and the Higgs boson play a sizeable role. For separating the squared $\mathcal{CP}$-even and $\mathcal{CP}$-odd amplitudes, the transverse momenta are most decisive.

We also investigated the model dependence of our limits. In particular, we checked whether $\mathcal{CP}$-violating couplings in the VBF channel could mimic a $\mathcal{CP}$-violating Higgs–gluon interaction in our analysis. Our results show that our analysis — in particular the ggF2j-SR — is very robust to such a situation and therefore allows disentangling $\mathcal{CP}$-violating Higgs couplings to massive vector bosons and gluons.

Finally, we reinterpreted our expected limits in terms of limits on the $\mathcal{CP}$ character of the top-Yukawa coupling. This reinterpretation relies on the assumption that no coloured BSM particles contribute sizeably to Higgs production to gluon fusion, which is motivated by the current limits from direct searches for such particles. Comparing our projections to a global fit Higgs precision measurements using the Run-2 data of the LHC, we demonstrated the large potential of ggF2j production as a precision probe of the $\mathcal{CP}$ nature of the Higgs boson.

## Acknowledgments

We thank Alberto Carnelli, Frederic Deliot, Christian Grefe, Andrei Gritsan, Lena Herrmann, Anastasia Kotsokechagia, Andrew Pilkington, Tilman Plehn, Matthias Saimpert and Laurent Schoeffel for useful discussions.

**Funding information**  HB acknowledges support from the Alexander von Humboldt Foundation. EF and MM acknowledge support by the Deutsche Forschungsgemeinschaft (DFG, German Research Foundation) under Germany's Excellence Strategy – EXC-2123 "Quantum-Frontiers" – 390837967.

## A  Details of the event generation

The model file used for the event generation is a custom UFO model file [115, 116] which only contains the terms in 1, as well as an effective Higgs-photon coupling. The $H\gamma\gamma$ coupling is assumed to be SM-like throughout this study. It is implemented as an effective coupling to enable the generation of the ggF2j process at the tree level. With G $\equiv$ g, GA $\equiv$ $\gamma$, H $\equiv$ scalar and A $\equiv$ pseudoscalar, there are four effective couplings: QGGH, QGGA, QGAGAH and QGAGAA that are used to generate specific parts of the amplitude. Setting QGGH==1, QGAGAH==1 (QGGA==1, QGAGAA==1) implies the effective interaction of a scalar (pseudoscalar) Higgs boson with exactly one pair of gluons and one pair of photons. The parameters can be set during the event generation, as seen in the following. All events are generated with two hard jets at the leading order in QCD. The overall rates are scaled to the next-to-leading order (NLO) results by flat K-factors from [56]. As an additional cross-check of our results, we generated two inclusive NLO Higgs plus jets samples using the Higgs Characterization UFO model [108] for $\alpha^{Hgg} = 0$ and $\alpha^{Hgg} = 35°$.[10] Passing these samples through the analysis workflow trained with LO samples, we find the $\Delta\chi^2$ value between the two $\alpha^{Hgg}$ hypotheses to agree within $\sim 10\%$ if comparing the LO and inclusive NLO samples (with the NLO $\Delta\chi^2$ being lower). This difference is likely mainly due to the background–signal classifiers being trained with LO samples. Assuming the same selection efficiencies as for the LO samples, the $\Delta\chi^2$ values calculated using the LO samples and the inclusive NLO samples agree within $\sim 1\%$ indicating the robustness of our results.

The syntax for the event generation is "generate p p > a a j j QGGH==1 QGAGAH==1 QED=4 QCD=4" for the $\left|\mathcal{M}_{\text{even}}\right|^2$ term in ggF2j, where the restrictions on the QED and QCD magnitude are set to exclude di-Higgs production. The interference and $\left|\mathcal{M}_{\text{odd}}\right|^2$ terms are created by replacing "QGGH==1" by "QGGH^2==1 QGGA^2==1" and "QGGA==1",[11] respectively. For the background processes, the effective Higgs-gluon coupling has been disabled in the model file. The VBF background is generated via "generate p p > a a j j \$\$ w+ w- z QGAGAH==1" where heavy vector bosons are forbidden to appear in the s-channel to exclude contributions from the $VH$ background. The latter is in turn generated via "generate p p > z > a a j j QGAGAH==1; add process p p > w+ > a a j j QGAGAH==1; add process p p > w- > a a j j QGAGAH==1".

---

[10]While this model supports NLO event generation, it does not allow to generate the interference term separately. Therefore, we concentrate on two benchmark points for this cross-check.

[11]We explicitly avoid using MadGraph's built-in decay routines by directly simulating the $\gamma\gamma jj$ final state including off-shell effects. The quantum number syntax allows separating the Higgs amplitude from other contributions. Using decay routines would not allow simulating the interference term separately.

An additional data set introducing $\mathcal{CP}$ violation in the VBF channel, used for the analysis in 6, is generated using the SMEFTsim 3.0 package [152]. Specifically, the U35 model is used in the $m_W$-scheme and we set $c_{\Phi\tilde{W}} = 1\text{ TeV}^{-1}$ as the only non-zero BSM parameter of the model. Events are then generated via "generate p p > a a j j $$ w+ w- z /a QCD=0 NP=1 NP^2==1" and we apply an additional diagram filter to ensure that only VBF events are generated.

## B  Distributions of kinematic variables

16 shows the distributions of all variables used in the training of the classifiers (see 4.1), separately for the three different $\mathcal{CP}$ contributions to the squared ggF2j amplitude. These include as low-level observables the energy $E$, transverse momentum $p_T$ and pseudorapidity $\eta$ of the reconstructed final state particles, as well as some high-level observables. An exception to this is the azimuthal angle $\phi$ of the aforementioned objects as this does not exhibit any separation between the three contributions due to the rotational invariance of the production process.

## C  Training uncertainty

In 17, the classifier outputs for the two $\mathcal{CP}$ discriminants in the ggF2j-like kinematic region are shown as the mean of 100 individual training processes, with the standard deviation in each bin plotted as a shaded region. This allows us to estimate the uncertainty due to the classifier training. We observe that the largest uncertainties come from bins close to 0 and 1 in the $P(c_g^2)$ classifier. For the $P_+ - P_-$ classifier, the uncertainty associated with the training process is negligible. In the VBF-like region (see 18), the visible statistical fluctuations in the $|\mathcal{M}_{\text{even}}|^2$ and $|\mathcal{M}_{\text{odd}}|^2$ contributions arise due to the comparably low number of ggF2j events in this kinematic region.

It should be noted that, in the present study, the uncertainty associated with the training process does not result in an uncertainty on the extraction of the Higgs–gluon couplings. Each trained classifier represents a slightly different observable. Consequently, the best-performing classifier can be chosen without introducing any further uncertainty in the final limits on the Higgs–gluon coupling.

## D  Likelihood evaluation

As mentioned above, the output of the $\mathcal{CP}$ classifiers is used to construct different observables. These observables are represented as histograms filled by all events from the respective test data set. Since each bin corresponds to a number of events, which follow a Poisson distribution, the histograms can be used to construct a likelihood

$$L_X = \frac{e^{-\lambda_X} \lambda_X^n}{n!}. \tag{D.1}$$

Here, $\lambda_X$ is the expected number of events for a specific $\mathcal{CP}$ hypothesis, while $n$ corresponds to the observed number of events. We set $n = \lambda_{\text{SM}}$, which corresponds to a perfect agreement of the data with the SM hypothesis, in order to construct limits on the Higgs–gluon coupling.

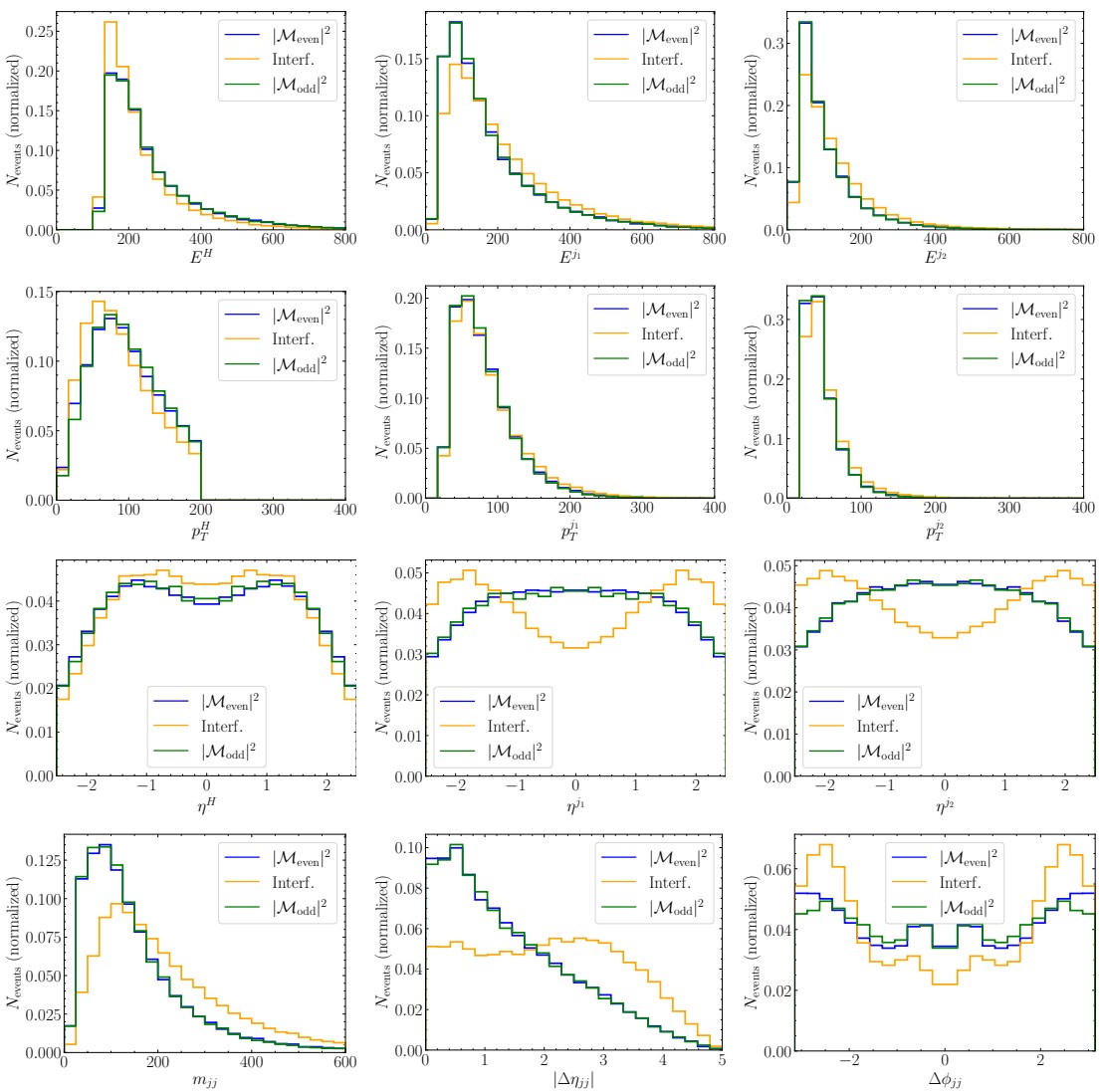

Figure 16: Distributions of the energy (first row), $p_T$ (second row) and $\eta$ (third row) of the Higgs boson (left column), as well as the jets leading (middle column) and sub-leading (right column) in $p_T$. The last row shows distributions of high-level observables, namely the invariant mass of the di-jet system $m_{jj}$ (left), as well as the difference in pseudorapidity $\Delta\eta_{jj}$ (middle) and azimuthal angle $\Delta\phi_{jj}$ (right) of the two jets. All distributions are split up into the $\left|\mathcal{M}_{\text{even}}\right|^2$ (blue), interference (orange) and $\left|\mathcal{M}_{\text{odd}}\right|^2$ (green) contributions to the total ggF2j cross section.

These limits are calculated using a binned likelihood ratio

$$
\begin{aligned}
t &= -2\ln\left(\frac{L_{\text{tot}}}{L_{\text{SM}}}\right) = -2\ln\left(\prod_i \frac{e^{-\lambda_{i,\text{tot}}}\lambda_{i,\text{tot}}^{n_i}}{e^{-\lambda_{i,\text{SM}}}\lambda_{i,\text{SM}}^{n_i}}\right) \\
&= -2\sum_i\left[\lambda_{\text{SM}}^i\cdot\ln\left(\frac{\lambda_{\text{tot}}^i}{\lambda_{\text{SM}}^i}\right) - \lambda_{\text{tot}}^i + \lambda_{\text{SM}}^i\right],
\end{aligned}
\tag{D.2}
$$

where $t$ is the test statistic and the sum runs over all bins $i$. $\lambda_{\text{SM}}^i = \lambda_{\text{E1}}^i + \lambda_{\text{BG}}^i$ is the expected SM distribution including ggF2j with $c_g = 1, \tilde{c}_g = 0$ and the BG processes, while $\lambda_{\text{tot}}^i = c_g^2\lambda_{\text{E1}}^i + c_g\tilde{c}_g\lambda_{\text{O}}^i + \tilde{c}_g^2\lambda_{\text{E2}}^i + \lambda_{\text{BG}}^i$ is the total bin value in the BSM case where $c_g$ and

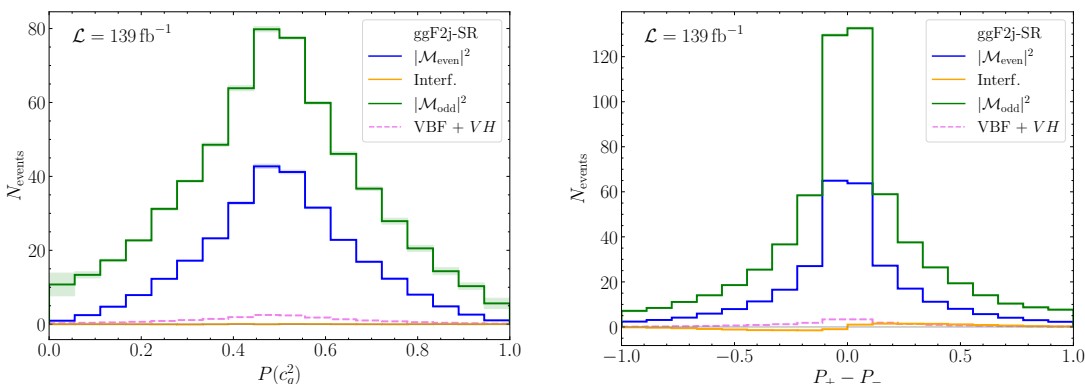

Figure 17: Distributions of the two $\mathcal{CP}$ discriminants for the mean of 100 trained classifiers in the ggF2j-like kinematic region.

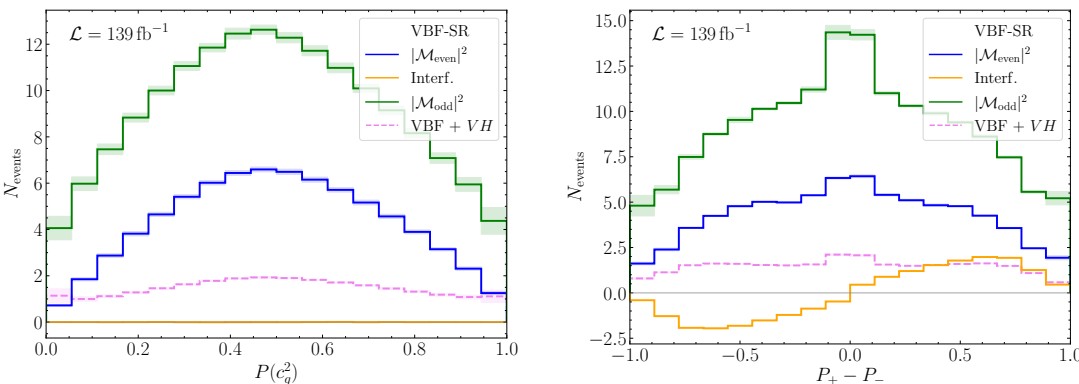

Figure 18: Distributions of the two $\mathcal{CP}$ discriminants for the mean of 100 trained classifiers in the VBF-like kinematic region.

$\tilde{c}_g$ can be varied. Following Wilks' theorem [153], we assume that $t$ follows a $\chi^2$-distribution and can, therefore, be used to construct confidence intervals.

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
