# Peer review of "Classifying the CP properties of the ggH coupling in H + 2j production"

_SciPost Physics Core, doi:SciPost Phys. Core 8, 006 (2025)_

## Round 1 · Referee Report · Chiara Mariotti (Referee 1) · 2024-1-4

Strengths
1.Interesting analysis that exploit a new final state, H+2jets, to study the CP properties of the ggH coupling. 2.Good introduction with a good description of the interplay of the different couplings, production modes and CP violation origin. 4. Interesting studies on the disentangling of the CP violation in the various couplings. 3.Use of Machine learning to improve the sensitivity and exploit many variables.
Weaknesses
- The paper is not very clear, it is hard to read. Sometime one find in the "Conclusions" the needed explanations.
- There is a big mistake: in Sec.3, it is written that the background can be subctrated and thus neglected (see later for an explanation).
- In section 3 there are sentences hard to read, i.e. or are wrong or I do not understand them (see later).
- There is no explanation on how the "interference events" are generated.
- The explanation on why the VBF-SR is considered is in the conclusion instead that in sec.4.
- It is confusing the treatment of Meven and Modd and the corresponding figures 4 and 5
- Figure 6 and 7 are not well explained and it is unclear how then one can use those distributions
- There is no definition of "limit" in sec.5.
- All the comparisons of the result presented in the paper with the experimental results are invalid, since the treatment (and statement) about the background and its subtraction is wrong.
Report
It can be an interesting signal-only study.
Requested changes
Section 3. 1. the second line mention that the ggH2j channel offers a unique sensitivity. I suggest to repeat the explanation of why (or at least point to the introduction). 2. line 8: The statement that background from "qq->qq+2" photons can be subctrated is correct only at the level of the "M_{gamma-gamma}" plot, when you want to measure the mass of the Higgs boson or its cross section. I.e. this background subtraction is valid only at statistical level. Event by event you cannot know if the event is signal or background, i.e. you cannot do a CP analysis background free ! I think anyway you can publish this analysis, but as an excercise on signal only (+VH+VBF background). As a consequence you cannot compare with other studies, expecially the one performed on data. 3. As a consequence of comment n.2, line 12 and 13 have to be deleted. 4. line 19: what does it mean "with particle having passed an electron, muon ... before."? 5. line 21: what does it mean : "Photons... are identified by observing electron...". It has no sense to me. 6. 4th paragraph - event generation: please explain how the interference sample is generated/constructed. (Since - to my knowledge- you cannot generate simply the interference term with madgraph, you should explain how you construct it). 7. 4th paragraph: the cut on the invariant mass of the 2 photons does not subctract all the non-Higgs background. Expecially in the gamma-gamma final state, the backgound is by far dominating. See any experimental plot from Atlas or CMS...
Section 4. 8. the procedure and the naming of the samples are quite confusing. The best would be to move the explanation on why you want to use the VBF-SR sample from the "conclusions" to this section, and clarify at the beginning the idea of the two SR. Otherwise it is not clear why the VBF, that is the background to ggH2j, is then considered signal.
Section 4.2 9. This part is very unclear to me: in the first bullet under figure 5 you talk about constructing a CP-even observable, but then while describing figure 6 you instead try to define a CP-odd observable. Figure 6 and 7 show P(c2) and P+-P- for Meven and Modd (and interference): Meven and Modd have the same shapes, only a different normalisation. How do you use these distribution to discriminate between Meven and Modd and get an estimate of P(c2) ?
Section 5 10. It is not clear the definition of LIMIT. Please define it. 11. why don't you try another CP hypothesys instead of only doing the study for the SM case ?
Section 5.3
12. line 5. add "full line" ---> ".... and the combined SRs (full line)"
13. 3rd paragraph: you cannot compare with experimental analyses since you are not considering the qcd background
Section 5.4 14. line 7. worth --> "worth" (under quotes "")
Section 7. 15 3rd and 4th paragraph: you cannot compare with experimental limit since you are not considering the background.
Conclusions: Again you cannot compare with other experimental results.

---

## Round 1 · Referee Report · Anonymous (Referee 2) · 2024-1-16

Strengths
the Higgs boson coupling to a top quark, or in an effective coupling to gluons.
The authors focus on the gluon fusion production of the Higgs boson in association
with two jets. While both experimental and theoretical ideas and techniques for
this analyses have already been established, the authors introduce one new aspect.
They suggest that in addition to the VBF-like topology used in the literature,
a distinct topology of events which do not look like VBF events can be employed.
Weaknesses
do not perform MC simulation of multiple jet radiation at higher orders in QCD
with careful matching to parton shower, do not consider all sources of systematc
uncertainties or detector effects, and do not properly account for possible CP
violating effects in the other Higgs boson production processes. This makes
numerical prediction of their studies unreliable. Nonetheless, the qualitative
suggestion to investigate a new topology of events in experimental analysis is
worth noting and to be taken for consideration by the ATLAS and CMS collaborations.
There is a possibility that further improvement of experimental results following
this suggestion will result from such a study, but this needs to be confirmed
by a more careful study.
Report
claiming significant improvements over the existing experimental results is
changed to a more careful assessment along the lines suggested above, and
comments below are taken into account.
More detailed comments are given below:
Page 3:
Second sentence: quantum numbers of the Higgs boson are mentioned, but ATLAS
and CMS references do not report tests of those.
References in general: Among the many ATLAS and CMS journal references, only one
actually quotes ATLAS Collaboration as the primary author (Ref.[47]). The suggestion
is to always refer to CMS or ATLAS Collaboration first, and then give the first
author and the rest of the journal information. Also suggest to use chronological
order in all groups of references.
Last sentence of the 2nd paragraph: It is not necessary for the Higgs boson to
be a mixed state in order to observe CP violation. In fact, even in the SM, the
purely CP-even Higgs boson could have CP violation in interactions due to three-loop
effects, though those are highly suppressed. The authors may want to refer to
CP violation in interactions instead.
The 4th paragraph assumes two distinct measurements: one using CP-odd observables,
and another using CP-even observables. However, in reality it is not always possible
to clearly separate the two types of observables, and the best measurement would always
come from using complete kinematic information using all observable information, CP-odd
and CP-even. Therefore, the whole concept of this paragraph is not clear. Instead,
the authors may want to introduce the idea of CP-odd and CP-even observables.
In the 5th paragraph: why use reference [8], but not [7]? Why start with Run1 ATLAS
reference [32], but not include the corresponding one from CMS?
page 4:
It is stated that "ggF production in association with one or two jets is more directly
sensitive." While this is correct for the two-jet configuration, the statement about the
one-jet configuration is not correct in general. Perhaps footnote 1 should be added here
already.
While an extensive list of references is given in the first paragraph [55,82-95], not all the
references deal with CP structure of the gluon fusion process. Some references deal primarily
with WVF, and in particular refs. [83-84] calculate CP-even background from gluon fusion.
At the same time, some of the original references for calculation of the CP-odd observables
are missing, in particular:
(1) for calculation of Delta\PhiJJ: https://arxiv.org/abs/hep-ph/0105325
(2) for calculation using matrix elements in gg->HJJ: https://arxiv.org/abs/1309.4819
Overall, references appear in a random order, so the suggestion is to order those chronologically
by submission date.
page 5: In discussion of Fig.1, fractions are quoted, e.g. 72% for gg initial state.
This is probably done in application to SM, but this is not clarified. However, Fig.1
shows non-SM contact interactions of the Higgs boson with gluons. Perhaps it should be
clarified that in this case those interactions are generated by SM loops, most likely
of the top quark.
Eq.(1): while this effective coupling describes the 2nd and 3rd diagrams in Fig.1,
the five-particle effective interaction from the 1st plot cannot be described by this Eq.
Suggest to re-draw the 1st diagram as a three-particle interaction, and leave the
five-particle interaction for separate consideration.
Eq.(4): as footnote-3 points, there are three CP-odd operators in the WBF process. As it has
been shown in Ref. https://arxiv.org/abs/2109.13363, two out of three operators with virtual
photons are less sensitive in this process. Therefore, one can consider one operator as the
primary one with the proper motivation.
Page 7: Any non-Higgs background is neglected in this analysis. While this might be Ok for
conceptual development of CP-odd observables and related studies, this does not provide a
realistic expectation in experimental analysis. An argument that this contribution to uncertainty
is small in the total event rate compared to other experimental uncertainties does not apply
CP asymmetry measurements because many rate uncertainties cancel in the asymmetries.
Appendix-A says that All events are generated at leading order and are scaled to NLO by flat
K-factors. This is not mentioned in the main body on page 7, and the aMC_NLO naming would
make everybody think that this generation is done at NLO. However, this brings a serious concern
about the associated jet modeling used in this paper. Pythia8 parton shower matching to LO
simulation is very unreliable. Even though up to two jets are generated at matrix element level,
additional jets may be modeled incorrectly and result in biased distributions when two jets are
selected for analysis. Perhaps the authors could quote the fraction of events when the two jets
selected are exactly those generated at matrix element level. This fraction could be substantially
smaller than 100% and this may indicate how well one can model such topology.
It is also not clear only two jets are modeled with MadGraph5_aMC@NLO , or if there is an inclusive
modeling of H+0, 1, 2jets with Pythia generating additional jets. Inclusive simulation is more
realistic.
Section 4: two classifies are employed: to separate gg2Fjj from background, and the other
to separate gg2Fjj from VBF. However, earlier it was stated that non-Higgs background was
neglected and only other Higgs production (VBF) was considered as background. The authors
need to clarify what they call backgrounds in this (and other) cases, and what exactly has
been simulated and what not.
The category of events ggF2j-SR has topology different from VBF and we expect the main
contribution to be from non-Higgs events. However, such events are neglected and it is not
clear how one can rely on any prediction of this analysis in regard to events in the ggF2j-SR
category.
Section 4.2: The training of CP classifies (CP even and CP odd) appears similar to that
summarized in Ref. https://arxiv.org/abs/2211.08353, section 2.1.3 in particular. This has
been worked out from first principles for matrix element calculations and then approximated
with machine learning approaches. Connection between the two allows justification of the method.
Fig.6: in the ggF2j-SR category, the main separation power in the shapes of observables
appears to be in tail of the green distribution (left side) of the CP-even observable (left),
which looks almost like a statistical fluctuation. This vision is supported by the large
uncertainty in the edge bins indicated in Appendix C. There is very little difference and
tiny interference contribution in the CP-odd observable (right). Combined with the lack of
background modeling in this category, this leads to a questionable prediction of separation
power in the ggF2j-SR category, which is shown in Section 5.
Section 5.3: comparison to experimental results is absolutely not fair: "In comparison
to the expected limits from existing experimental analyses [37, 38, 45], our combined
limit is significantly stronger." This is because experimental results take all effects
into account, while this work made many approximations and did not consider realistic
MC simulation and all sources of background, statistical, and systematic uncertainties.
The authors of experimental analyses did consider many effects discussed in this paper
and did not find such an improvement in separation power after considering all effects
which degrade performance.
Section 5.4: analysis of separation power of individual observables is somewhat random
since the choice of observables is also random. The key point in choosing input observables
is to represent the full kinematic information and the rest is an arbitrary choice of
the authors. With the present choice, some pattern can be observed, but it is mostly
limited to the delta\Phi_JJ observable which is already known in the literature.
Section 6: the whole section appears to present biased results and wrong picture.
This is especially evident from the statement that the "better" constraints in the
VBF-SR on cg and ~cg are faked from CP violation in the HVV couplings. It is not
possible to obtain better constraints when additional CP violating effects are allowed.
The results can become only worse with additional free parameters if statistical
interpretation is correct. The correct approach would be the following: The model
is extended to allow an additional CP-violalting parameter in the VBF+VH process.
When this parameter is set to zero, old results are recovered. When this parameter
is allowed to float, fits are performed on the same samples as before. The resulting
constraints on cg and ~cg will become weaker when the new CP-odd parameter is floated
in addition to the other parameters. The critical question is how much weaker the
constraints become.
Section 7: comparison to experimental results following Eq.(5) are not fair,
as discussed above. Many effects are neglected in the present analysis, and one
cannot rely on numerical prediction for comparison to the solid experimental
measurements. The same comment applies to Table 2.

---

## Round 2 · Referee Report · Anonymous (Referee 1) · 2024-7-25

Strengths

1.Interesting analysis that exploit a new final state, H+2jets, to study the CP properties of the ggH coupling. 2.Good introduction with a good description of the interplay of the different couplings, production modes and CP violation origin. 4. Interesting studies on the disentangling of the CP violation in the various couplings. 3.Use of Machine learning to improve the sensitivity and exploit many variables.

Weaknesses

  1. The paper is a "signal only" study: it considers only the Higgs boson produced in different modes, but not any other background. The background is dominating in the "gamma-gamma" final state.

Report

The paper presents a new channel to study the CP properties of the Higgs boson.
It is a preliminary study, since it does not consider the background.
It deserves to be published.

Recommendation

Publish (meets expectations and criteria for this Journal)

---

## Round 2 · Referee Report · Anonymous (Referee 2) · 2024-12-2

Report

The authors have revised the manuscript and toned down their comparison to realistic experimental measurements. However, throughout the paper, they continue to mistakenly refer to the missing effect solely as "systematic uncertainties" and refer to expected "significant improvements."

For example, in the Conclusion the authors continue to assert a significant improvement: "These results suggest that significant improvements in current experimental limits are possible with well-established techniques. We leave a more detailed study including the effects of systematic uncertainties for future work." It is not just systematic uncertainties that have been overlooked, but also statistical uncertainties arising from the neglect of the dominant non-Higgs backgrounds. Therefore, "significant improvements" have not been demonstrated.

Another statement appears at the end of Section 5.3: "It should be noted, however, that the limits derived here are expected to degrade once systematic uncertainties, particularly the non-Higgs background, are taken into account. We leave a full investigation of all systematic effects for further work." Once again, the non-Higgs background is not only a source of systematic uncertainties but also the main contributor to the statistical uncertainties in the analysis of Higgs decay to two photons.

The recommendation is to adopt more cautious language in the Abstract and in the Conclusion, and substitute the sentence "Our results suggest that significant improvements in current experimental limits are possible" with something like "These studies indicate that further improvements in the current experimental limits could be achievable using our techniques." Which can be elaborated further in the Conclusion: "The extent of these improvements will depend on factors such as the realistic treatment of non-Higgs backgrounds and systematic uncertainties, which are left for future work."

Finally, in Footnote 3 on page 7, the author now state: "In this work, we concentrate on the O_{\Phi W\tilde} which has the largest impact on VBF production [110]."However, the conclusion of Ref. [110] differs, as largest impact arises from an operator that is a linear combination of O_{\Phi W\tilde}, O_{\Phi B\tilde}, and O_{\Phi BW\tilde}.

With the above suggestions, I recommend the manuscript for publication.

Recommendation

Ask for minor revision

---

## Round 2 · Author Response

Dear Editor and Referees,

We thank the referees for the constructive and helpful remarks. We have addressed the various comments raised in the referees’ report and modified the text accordingly. In the list of changes, we answer in detail to the different points raised by the referees. We hope that with these changes, our article can now be accepted for publication in its present form.

Sincerely,
H. Bahl, M. Hannig, M. Menen, E. Fuchs

---

## Round 2 · List of Changes

Referee 1:

  1. Section 3. the second line mention that the ggH2j channel offers a unique sensitivity. I suggest to repeat the explanation of why (or at least point to the introduction).

We added an explanatory sentence.

  1. line 8: The statement that background from "qq->qq+2" photons can be subctrated is correct only at the level of the "M_{gamma-gamma}" plot, when you want to measure the mass of the Higgs boson or its cross section. I.e. this background subtraction is valid only at statistical level. Event by event you cannot know if the event is signal or background, i.e. you cannot do a CP analysis background free ! I think anyway you can publish this analysis, but as an excercise on signal only (+VH+VBF background). As a consequence you cannot compare with other studies, expecially the one performed on data.

We removed the paragraph stating that the non-Higgs background is easily subtractable and instead commented on the limitations of our study.

  1. As a consequence of comment n.2, line 12 and 13 have to be deleted.

See comment above.

  1. line 19: what does it mean "with particle having passed an electron, muon ... before."?

We reformulated the sentence.

  1. line 21: what does it mean : "Photons... are identified by observing electron...". It has no sense to me.

We clarified the sentence.

  1. 4th paragraph - event generation: please explain how the interference sample is generated/constructed. (Since - to my knowledge- you cannot generate simply the interference term with madgraph, you should explain how you construct it).

We moved the reference to Appendix A to the end of this paragraph. Moreover, we added to footnote in the Appendix explaining that the used syntax allows us to bypass the limitation of MadGraph encountered when generating interference events if using the decay chain syntax.

  1. 4th paragraph: the cut on the invariant mass of the 2 photons does not subctract all the non-Higgs background. Expecially in the gamma-gamma final state, the backgound is by far dominating. See any experimental plot from Atlas or CMS...

We agree and modified the sentence accordingly.

  1. Section 4. the procedure and the naming of the samples are quite confusing. The best would be to move the explanation on why you want to use the VBF-SR sample from the "conclusions" to this section, and clarify at the beginning the idea of the two SR. Otherwise it is not clear why the VBF, that is the background to ggH2j, is then considered signal.

We modified the beginning of Section 4 to clarify the meaning of the two SRs and that ggF2j is always our signal process.

  1. Section 4.2 This part is very unclear to me: in the first bullet under figure 5 you talk about constructing a CP-even observable, but then while describing figure 6 you instead try to define a CP-odd observable. Figure 6 and 7 show P(c2) and P+-P- for Meven and Modd (and interference): Meven and Modd have the same shapes, only a different normalisation. How do you use these distribution to discriminate between Meven and Modd and get an estimate of P(c2) ?

We extended the text to address these points and made it more clear what the task of the different classifiers is. While the P (c2g) distributions for the Meven and Modd contributions have a similar trend in the shape, there are larger differences which are visible more clearly if the same normalisation is applied. We have extended the text to include a more accurate description of this.

  1. Section 5 It is not clear the definition of LIMIT. Please define it.

We clarified that we perform a parameter fit assuming SM data.

  1. why don’t you try another CP hypothesys instead of only doing the study for the SM case?

We agree that this would be an interesting application and added a plot to Fig. 10, as well as a discussion in Sec. 5.3. The plot projecting to higher luminosities was moved to a newly created Fig. 11.

  1. Section 5.3 line 5. add "full line" —> ".... and the combined SRs (full line)"

The blue dashed and the black dashed lines overlap such that it looks like a solid line. This is now mentioned in a new footnote.

  1. 3rd paragraph: you cannot compare with experimental analyses since you are not considering the qcd background

We removed the comparison.

  1. Section 5.4 line 7. worth –> "worth" (under quotes "")

We implemented this change.

  1. Section 7. 3rd and 4th paragraph: you cannot compare with experimental limit since you are not considering the background.

We removed the comparison.

Referee 2:

  1. Page 3: Second sentence: quantum numbers of the Higgs boson are mentioned, but ATLAS and CMS references do not report tests of those.

We added references to the quantum number measurements to our statement.

  1. References in general: Among the many ATLAS and CMS journal references, only one actually quotes ATLAS Collaboration as the primary author (Ref.[47]). The suggestion is to always refer to CMS or ATLAS Collaboration first, and then give the first author and the rest of the journal information. Also suggest to use chronological order in all groups of references.

We have modified our references so that the respective collaboration is always quoted as the primary author. We checked also that our groups of references are ordered chronologically, unless they include papers that were already cited earlier in the text.

  1. Last sentence of the 2nd paragraph: It is not necessary for the Higgs boson to be a mixed state in order to observe CP violation. In fact, even in the SM, the purely CP-even Higgs boson could have CP violation in interactions due to three-loop effects, though those are highly suppressed. The authors may want to refer to CP violation in interactions instead.

We agree and modified the statement accordingly.

  1. The 4th paragraph assumes two distinct measurements: one using CP-odd observables, and another using CP-even observables. However, in reality it is not always possible to clearly separate the two types of observables, and the best measurement would always come from using complete kinematic information using all observable information, CP-odd and CP-even. Therefore, the whole concept of this paragraph is not clear. Instead, the authors may want to introduce the idea of CP-odd and CP-even observables.

We adjusted this paragraph.

  1. In the 5th paragraph: why use reference [8], but not [7]? Why start with Run1 ATLAS reference [32], but not include the corresponding one from CMS?

We have added the corresponding CMS analysis (prior Ref [7], now Ref [3]) to the list of references. We also added the CMS reference for Run1.

  1. page 4: It is stated that "ggF production in association with one or two jets is more directly sensitive." While this is correct for the two-jet configuration, the statement about the one-jet configuration is not correct in general. Perhaps footnote 1 should be added here already.

We agree and moved the footnote to the corresponding mention of the ggF + one jet process.

  1. While an extensive list of references is given in the first paragraph [55,82-95], not all the references deal with CP structure of the gluon fusion process. Some references deal primarily with WVF, and in particular refs. [83-84] calculate CP-even background from gluon fusion. At the same time, some of the original references for calculation of the CP-odd observables are missing, in particular: (1) for calculation of DeltaPhiJJ: https://arxiv.org/abs/hep-ph/0105325 (2) for calculation using matrix elements in gg->HJJ: https://arxiv.org/abs/1309.4819 Overall, references appear in a random order, so the suggestion is to order those chronologically by submission date.

We have added the two mentioned references and have moved two of the references which did not deal with CP violation to a more fitting place in the paper (when discussing the heavy top limit). As stated above, we checked that all references are ordered chronologically unless they appear earlier in the paper.

  1. page 5: In discussion of Fig.1, fractions are quoted, e.g. 72% for gg initial state. This is probably done in application to SM, but this is not clarified. However, Fig.1 shows non-SM contact interactions of the Higgs boson with gluons. Perhaps it should be clarified that in this case those interactions are generated by SM loops, most likely of the top quark.

We have added some information to make the discussion more clear.

  1. Eq.(1): while this effective coupling describes the 2nd and 3rd diagrams in Fig.1, the five-particle effective interaction from the 1st plot cannot be described by this Eq. Suggest to re-draw the 1st diagram as a three-particle interaction, and leave the five-particle interaction for separate consideration.

We exchanged the plot with the three-particle interaction.

  1. Eq.(4): as footnote-3 points, there are three CP-odd operators in the WBF process. As it has been shown in Ref. https://arxiv.org/abs/2109.13363, two out of three operators with virtual photons are less sensitive in this process. Therefore, one can consider one operator as the primary one with the proper motivation.

We agree and modified footnote 3 accordingly.

  1. Page 7: Any non-Higgs background is neglected in this analysis. While this might be Ok for conceptual development of CP-odd observables and related studies, this does not provide a realistic expectation in experimental analysis. An argument that this contribution to uncertainty is small in the total event rate compared to other experimental uncertainties does not apply CP asymmetry measurements because many rate uncertainties cancel in the asymmetries.

We have removed our statement that the uncertainty of the non-Higgs background is small and instead left a comment about the limitations of our study.

  1. Appendix-A says that All events are generated at leading order and are scaled to NLO by flat K-factors. This is not mentioned in the main body on page 7, and the aMC_NLO naming would make everybody think that this generation is done at NLO. However, this brings a serious concern about the associated jet modeling used in this paper. Pythia8 parton shower matching to LO simulation is very unreliable. Even though up to two jets are generated at matrix element level, additional jets may be modeled incorrectly and result in biased distributions when two jets are selected for analysis. Perhaps the authors could quote the fraction of events when the two jets selected are exactly those generated at matrix element level. This fraction could be substantially smaller than 100% and this may indicate how well one can model such topology.

We moved the information about the LO event generation from the appendix to the main body of the text for clarity. In order to test the robustness of our results we generated an inclusive sample of ggF events with one and two additional jets at NLO for angles of α = 0◦, 35◦. We then passed our inclusive NLO events through the same cutflow and classifiers as the exclusive LO events. From the obtained distributions we calculated the χ2 value for excluding the 35◦ hypothesis given a SM-like measurement and obtained similar results for both LO and NLO samples. There was only a slight tendency towards lower χ2 values in the ggF2j SR and towards higher χ2 values in the VBF SR which is likely due to the signal-background classifier being trained on LO samples. This information is now provided in Appendix A.

  1. It is also not clear only two jets are modeled with MadGraph5_aMC@NLO , or if there is an inclusive modeling of H+0, 1, 2jets with Pythia generating additional jets. Inclusive simulation is more realistic.

In this study, we only simulated events with two jets at parton level. This information has been added to the appendix. We have performed a test with an inclusive NLO sample as described above and will keep the inclusive simulation in mind for future studies.

  1. Section 4: two classifiers are employed: to separate gg2Fjj from background, and the other to separate gg2Fjj from VBF. However, earlier it was stated that non-Higgs background was neglected and only other Higgs production (VBF) was considered as background. The authors need to clarify what they call backgrounds in this (and other) cases, and what exactly has been simulated and what not.

We have modified the section to explain in more detail which processes the classifiers are separating and that we are not considering any non-Higgs backgrounds in our study.

  1. The category of events ggF2j-SR has topology different from VBF and we expect the main contribution to be from non-Higgs events. However, such events are neglected and it is not clear how one can rely on any prediction of this analysis in regard to events in the ggF2j-SR category.

While we do not consider the non-Higgs background in our study, we have added a comment that the non-Higgs background are expected to primarily land in the ggF2j-SR. We further removed comments comparing our results to experimental analyses for this reason.

  1. Section 4.2: The training of CP classifies (CP even and CP odd) appears similar to that summarized in Ref. https://arxiv.org/abs/2211.08353, section 2.1.3 in particular. This has been worked out from first principles for matrix element calculations and then approximated with machine learning approaches. Connection between the two allows justification of the method.

We agree that the work mentioned by the referee is closely related to our approach. We now mention it in Section 4.2 (below the bullet points).

  1. Fig.6: in the ggF2j-SR category, the main separation power in the shapes of observables appears to be in tail of the green distribution (left side) of the CP-even observable (left), which looks almost like a statistical fluctuation. This vision is supported by the large uncertainty in the edge bins indicated in Appendix C. There is very little difference and tiny interference contribution in the CP-odd observable (right). Combined with the lack of background modeling in this category, this leads to a questionable prediction of separation power in the ggF2j-SR category, which is shown in Section 5.

The two distributions in Fig. 6 show a much more distinct difference in their shapes when the same normalization is applied to both of them. For the fluctuation in the tail, we have checked that the amount of data which was used to train our classifiers is large enough so that the MC statistical uncertainty is much smaller than the Poisson uncertainty expected from the distributions (without taking any background into account). Instead, the fluctuation in the tails of the mentioned distributions come from an uncertainty in the training of the classifier. This is, however, not the sole reason for the separation power and mainly allows us to choose the classifier resulting in the best separation. We have clarified this in the text.

  1. Section 5.3: comparison to experimental results is absolutely not fair: "In comparison to the expected limits from existing experimental analyses [37, 38, 45], our combined limit is significantly stronger." This is because experimental results take all effects into account, while this work made many approximations and did not consider realistic MC simulation and all sources of background, statistical, and systematic uncertainties. The authors of experimental analyses did consider many effects discussed in this paper and did not find such an improvement in separation power after considering all effects which degrade performance.

We agree and consequently have removed the direct comparison with experimental studies.

  1. Section 5.4: analysis of separation power of individual observables is somewhat random since the choice of observables is also random. The key point in choosing input observables is to represent the full kinematic information and the rest is an arbitrary choice of the authors. With the present choice, some pattern can be observed, but it is mostly limited to the deltaPhi_JJ observable which is already known in the literature.

For the training of our classifiers, we chose in addition to the lowest level kinematic information only variables which are commonly used in analyses and deliberately not took into account more complicated observables (such as e.g. the bi observables). The point of this section was not to make a study on which higher-level observable is the best, but to shine some light on how the classifier achieves its separation.

  1. Section 6: the whole section appears to present biased results and wrong picture. This is especially evident from the statement that the "better" constraints in the VBF-SR on cg and cgt are faked from CP violation in the HVV couplings. It is not possible to obtain better constraints when additional CP violating effects are allowed. The results can become only worse with additional free parameters if statistical interpretation is correct. The correct approach would be the following: The model is extended to allow an additional CP-violating parameter in the VBF+VH process. When this parameter is set to zero, old results are recovered. When this parameter is allowed to float, fits are performed on the same samples as before. The resulting constraints on cg and cg will become weaker when the new CP-odd parameter is floated in addition to the other parameters. The critical question is how much weaker the constraints become.

We agree with the comments of the referee and have modified our analysis accordingly. We now float cPhiW in our analysis and profile over it to obtain limits on (cg, cg~). Our conclusions regarding the impact of CP-violating HVV couplings did not change.

  1. Section 7: comparison to experimental results following Eq.(5) are not fair, as discussed above. Many effects are neglected in the present analysis, and one cannot rely on numerical prediction for comparison to the solid experimental measurements. The same comment applies to Table 2.

We removed the comparisons and Table 2

---

## Round 3 · Author Response

Dear Editor, dear Referees,

we thank you for the constructive feedback. We have implemented the suggestions and hope that our article is now suitable for publication.

Sincerely,

H. Bahl, E. Fuchs, M. Hannig, M. Menen

---

## Round 3 · List of Changes

• In the abstract, Section 5.3 and the conclusion, we weakened the statements about "significant improvements in experimental measurements" and emphasized that there are not just systematic uncertainties missing in our study.
  • We modified footnote 3 according to the suggestion.

---

## Editorial Decision

published